# Binding of regulatory proteins to nucleosomes is modulated by dynamic histone tails

Yunhui Peng [1], Shuxiang Li [2], Alexey Onufriev [3,4,5], David Landsman [1] & Anna R. Panchenko [2✉]

Little is known about the roles of histone tails in modulating nucleosomal DNA accessibility and its recognition by other macromolecules. Here we generate extensive atomic level conformational ensembles of histone tails in the context of the full nucleosome, totaling 65 microseconds of molecular dynamics simulations. We observe rapid conformational transitions between tail bound and unbound states, and characterize kinetic and thermodynamic properties of histone tail-DNA interactions. Different histone types exhibit distinct binding modes to specific DNA regions. Using a comprehensive set of experimental nucleosome complexes, we find that the majority of them target mutually exclusive regions with histone tails on nucleosomal/linker DNA around the super-helical locations ± 1, ± 2, and ± 7, and histone tails H3 and H4 contribute most to this process. These findings are explained within competitive binding and tail displacement models. Finally, we demonstrate the crosstalk between different histone tail post-translational modifications and mutations; those which change charge, suppress tail-DNA interactions and enhance histone tail dynamics and DNA accessibility.

[1] National Center for Biotechnology Information, National Institutes of Health, Bethesda, MD, USA. [2] Department of Pathology and Molecular Medicine, School of Medicine, Queen's University, Kingston, ON, Canada. [3] Physics Department, Virginia Tech, VA, USA. [4] Computer Science Department, Virginia Tech, VA, USA. [5] Center for Soft Matter and Biological Physics, Virginia Tech, VA, USA. ✉email: anna.panchenko@queensu.ca

In eukaryotic cells, DNA is packaged in the form of chromatin and should be dynamically accessed during transcription and replication processes with high spatiotemporal precision. These seemingly contradictory tasks of DNA packaging and DNA access have been of tremendous research interest. Nucleosomes represent the basic subunits of chromatin structure and comprise a histone octamer of four types of core histones, two copies each (H2A, H2B, H3, and H4) and ~147 bp of DNA wrapped around them[1]. Intrinsically disordered histone tails flanking histone core domains play particularly important roles, and experiments show that deletions of histone tails may result in the transient unwrapping of DNA, an increase in the nucleosome sliding rate, and a decrease in nucleosome stability[2–4]. Moreover, histone tails may contribute to the inter-nucleosomal interactions and affect the higher-order chromatin structure[5–7].

Histone tails have a high degree of conformational flexibility and might protrude into the solvent and remain perpetually accessible for binding by chromatin factors[1,8–10]. However, there is growing evidence that histone tails can extensively interact with the nucleosomal and linker DNA[11–19], which raises the possibility that tails may modulate the nucleosomal and linker DNA accessibility and regulate the nucleosome recognition by binding partners. It has been shown that despite the lower net negative charge of the nucleosome compared to the free DNA, nucleosomes are characterized by an enhanced negative charge density (so-called electrostatic focusing) even with the intact positively charged histone tails[20]. However, there are very few studies systematically characterizing the histone tail conformational ensemble in the context of the full nucleosome, physicochemical properties of their binding to DNA, and their functional roles in regulatory mechanisms[11,12,21,22]. This is explained by the difficulty in experimentally observing and simulating the intrinsically disordered tails' conformational space in the context of the full nucleosome.

Here we explore a spectrum of conformational states of disordered histone tails in the context of the full nucleosome to understand how conformational dynamics of histone tails may modulate the DNA solvent accessibility and the recognition of nucleosome-binding partners. We perform extensive sampling of tail conformations totaling 65 microseconds simulated trajectories. We find rapid interconversions between histone tail–DNA bound and unbound states and show that the ensemble of tail conformations adheres to the nucleosome two-fold symmetry requirement and provides reasonable estimates of tail–DNA dissociation constants. Finally, we utilize experimental data on nucleosome structural complexes and dissociation constants of various chromatin factors in order to explore how tail dynamics may affect the interactions of nucleosomes with their binding partners. We find that many chromatin factors and histone tails target overlapping and mutually exclusive regions on nucleosomal or linker DNA, pointing to generalized competitive binding or tail displacement mechanisms in nucleosome recognition by binding partners. Our study further demonstrates that post-translational modifications (PTMs) and mutations in histone tails can alter the tail–DNA binding modes and regulate the binding of partners to the nucleosome.

## Results

**DNA binding properties differ between histone tail types.** Histone tails have high conformational flexibility, and their conformational sampling represents a major challenge. To address this problem, we have built four nucleosome models with different initial histone tail configurations and performed 42 different runs totaling about 41 μs of simulations of unmodified tails (Supplementary Table 1) and 24 μs simulations of tails with PTMs or mutations (Supplementary Table 2), which can provide a quite extensive overview of the histone tails' conformational and interaction landscape. In concordance with other in silico and experimental studies[11,12,14,23,24], we observe a relatively rapid condensation and extensive interactions of histone tails with the nucleosomal and linker DNA. Our simulations using the OPC water model show many rapid interconversions between tail–DNA bound and unbound states (Fig. 1), pointing to a more dynamic histone tail behavior compared to simulations with the TIP3P water model (short simulations) where histone tails remain in the bound state with DNA most of the time (Supplementary Note 1)[11,18]. Thereafter throughout the paper, we only report the results of simulations with the OPC water model.

To further characterize the kinetics of histone tail–DNA interactions, we count a total number of transitions from unbound to bound states and compute histone tail residence time to estimate the effective time that histone tails stay bound to the DNA molecule (as the inverse of the dissociation rate constant, $\tau = 1/k_{\text{off}}$), evaluating full tail residence time ($\tau_{\text{f}}$) and individual residue residence time ($\tau_{\text{r}}$). As can be seen in Fig. 1b–d, the number of binding-unbinding events, residence time, and binding free energies vary considerably between histone types. Histone H3 has the longest residence time among all tails, up to five microseconds, and is characterized by relatively higher binding free energy and fewer unbinding events. It is followed by H2B, H4, and H2A N-terminal tails, whereas H2A C-terminal tails have the shortest average residence time and lowest binding free energy with DNA compared to other tails (Fig. 1c, d, Supplementary Table 4). ANOVA analysis and Tukey HSD test confirm that H3 and H2A-C terminal tails have significantly different residence times compared to other tails (Supplementary Table 6). To characterize the binding kinetics in more detail, we calculate the individual residue residence time ($\tau_{\text{r}}$) (Supplementary Fig. 3), which is found to be on the time scale of several to tens of nanoseconds, demonstrating very rapid and frequent transitions between bound and unbound states of each histone tail residue and jittery conformational rearrangements of histone tails in the bound state. Congruent with these findings, residues with long $\tau_{\text{r}}$ have a high binding free energy with DNA (Supplementary Fig. 3). We further compare our estimates of binding free energies from histone tail conformational ensemble statistics with the binding free energy estimates coming from a set of independent MM/PBSA calculations (Supplementary Tables 4 and 5). Overall, we observe a strong linear association between the histone tail–DNA binding free energies derived from the tail conformational ensemble statistics and MM/GBSA calculations for different values of cut-off parameters (Supplementary Fig. 4). We further analyze the secondary structure content of histone tails and observe that histone tails remained highly unstructured in simulations with the exception of the transient alpha-helical formation on H3 tail (Supplementary Fig. 5),

As was observed in previous studies, one of the most prevalent modes of interactions between histone tails and DNA was the insertion of the arginine and, in some cases, lysine side chains into the DNA minor and major grooves serving as anchors stabilizing these interactions[12,25,26]. Supplementary Fig. 3 shows that anchoring of certain arginines is critical in determining the tail's longest residence time. H2A C-terminal tails are the shortest tails, which do not have arginine residues and exhibit the shortest $\tau_{\text{f}}$, while H3 tails have the longest tail length, are arginine-rich, and have the longest $\tau_{\text{f}}$. For tails without arginine residues, the most prominent mode of interaction is between lysine and serine residues and DNA.

**Histone tail dynamics modulate the nucleosomal and linker DNA accessibility.** Interactions between histone tails and DNA may decrease their respective solvent accessibility. At the same time, upon

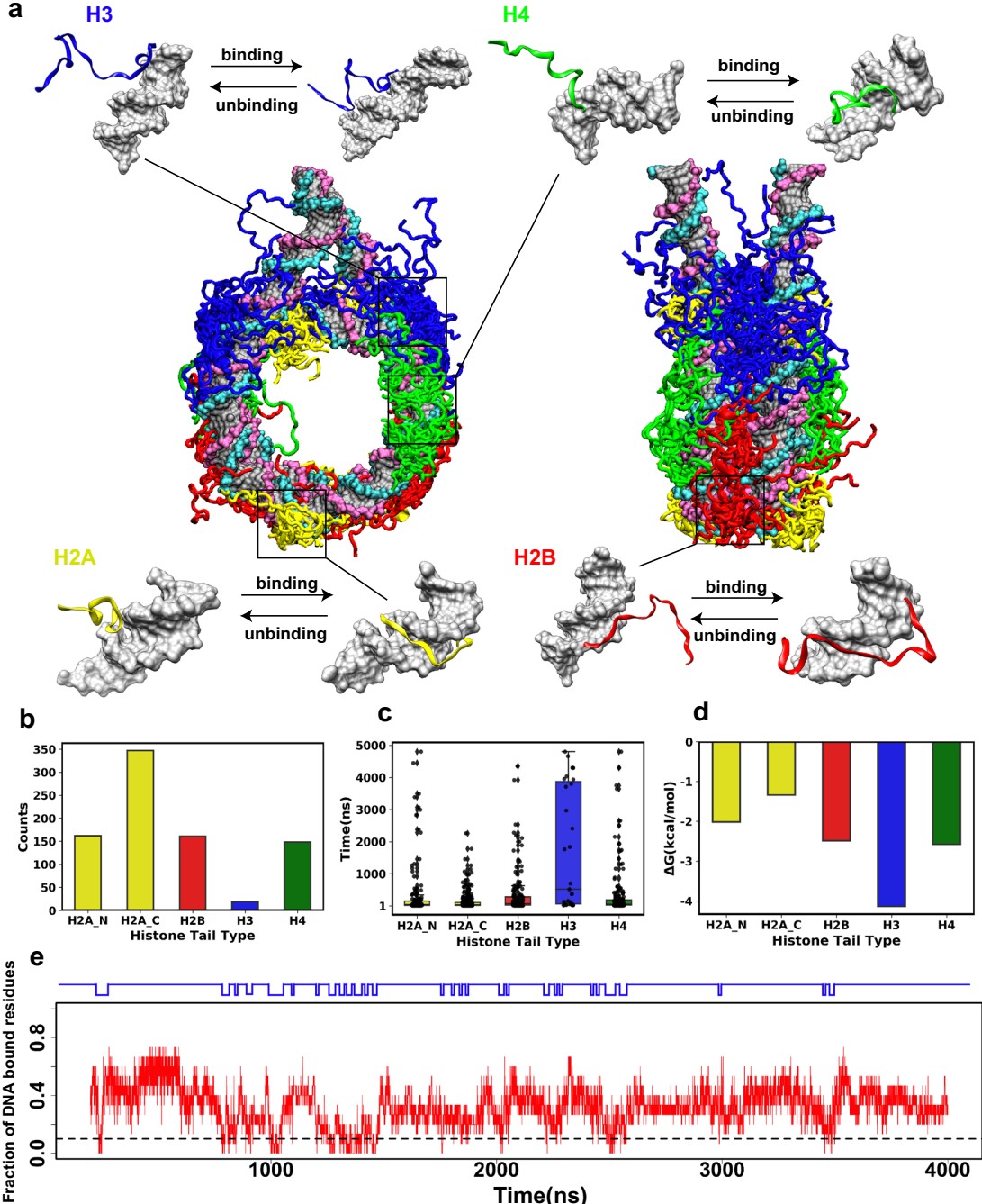

**Fig. 1 Binding of histone tails to nucleosomal and linker DNA in the context of the full nucleosome. a** Interconversions of DNA-bound and unbound tail conformations. The conformational snapshots are taken from the last frame of each simulation run and superimposed onto the initial models by minimizing RMSD values of $C^\alpha$ atoms in histone cores. While we observe multiple binding/unbinding events in the simulations, only a few snapshots are shown for clarity. **b** A total number of full histone tail binding/unbinding events observed in all simulations for both copies of histones. **c** Full histone tail residence time. Each point represents a binding/unbinding event observed in simulations for each histone copy ($n$(H2A_N) = 174, $n$(H2A_C) = 359, $n$(H2B) = 173, $n$(H3) = 31, $n$(H4) = 160). Data points with residence time shorter than 10 ns are excluded as this time is required for establishing stable interactions with DNA. An unbound state for the full tail is defined if no more than 10% of the tail residues maintain contacts with DNA (other cut-off values are given in Supplementary Fig. 2). Box-plot elements are defined as: center line, median; box limits, upper and lower quartiles; whiskers are drawn at values equal to 1.5× interquartile range; **d** The histone tail–DNA standard binding free energy derived from counting the number of bound/unbound events in histone tail conformational ensemble in all simulations for both copies of histones (number of binding/unbinding events ≥ 5, Supplementary Table 4). **e** A representative run shows a fraction of DNA bound residues and tail binding/unbinding events during simulation for H2B tails (see Supplementary Figs. 6–10 for other tails). Source data are provided as a Source Data file.

unbinding, histone tails and DNA become more accessible to nucleosome-binding proteins[11]. We analyze the interaction modes of histone tails and estimate the changes of nucleosomal and linker DNA solvent accessibility imposed by the tail binding. Due to the 2-fold pseudo-symmetry of the nucleosome structure, upon exhaustive conformational sampling, one should expect that each histone copy samples a similar phase space region (Supplementary Fig. 11). Indeed, we show that there is a statistically significant correlation between the mean number of tail–DNA contacts occupied by each copy of histone tails (Supplementary Fig. 12). To further assess the convergence of different simulation runs, we compare the histone tail-binding site locations between simulations starting from different initial configurations and they show the convergence on similar tail–DNA binding sites (Supplementary Table 7 and Supplementary Fig. 13). Therefore, below we report a combined conformational ensemble from both copies of histone tails. The correlation coefficients increase in values as the simulations progress.

As can be seen in Fig. 2a, b, tails of different histone types preferably interact with the specific DNA regions. H2A N-terminal tails bind to the nucleosomal DNA at superhelical locations (SHL) ± 4, whereas H2A C-terminal tails are mostly bound at SHL ± 7 and near the dyad. Interaction modes of H2B and H4 tails encompass a more extensive DNA-binding interface compared to other tails due to their dynamic behavior, allowing H2B and H4 tails to search a large surface area on DNA without being kinetically trapped in specific conformations. Being the longest, H3 tails can also interact with DNA in multiple regions with the longest residence time: near the dyad, at SHL ± 6 to ± 7 as well as with the linker DNA.

Binding of histone tails can partially or substantially occlude specific DNA regions from the solvent (Fig. 2). Some DNA regions that interact with histone tails undergo a substantial decrease of solvent accessible surface area (SASA) up to 100 Å² (Fig. 2c). DNA regions around SHL ± 4 undergo the most extensive accessibility changes, with at least 20% of SASA decrease in more than 70% MD frames followed by the DNA locations SHL ± 1, ±2, and ±7. The change of the DNA SASA is also highly correlated with the number of contacts between DNA and tails (Fig. 2a, c; Supplementary Fig. 14).

**Histone tails and nucleosome-binding proteins target overlapping regions on nucleosomal/linker DNA.** Nucleosomes, being the hubs in epigenetic signaling pathways, are targeted by a wide spectrum of nucleosome-binding proteins that interact with the specific regions on nucleosomal/linker DNA and histones[27–29]. To this end, we perform a systematic analysis of interaction modes of nucleosome-binding proteins using available nucleosome complex structures in PDB[30], totaling 131 structures (Fig. 3a). The functional classification of nucleosome-binding proteins shows that the majority of them include chromatin remodelers and transcription regulatory proteins. 86 nucleosome-binding partners recognize some part of DNA molecules, and most of them exhibit multivalent binding modes interacting with both histones and DNA. Among multivalent interactors, about 60% of them recognize histone tails as well as DNA (H2A-C, H3, and H4 tails, Supplementary Table 8), and the rest recognize DNA and histone core residues. An example of chromatin remodeler ISWI, which binds to nucleosomal DNA at SHL ± 2

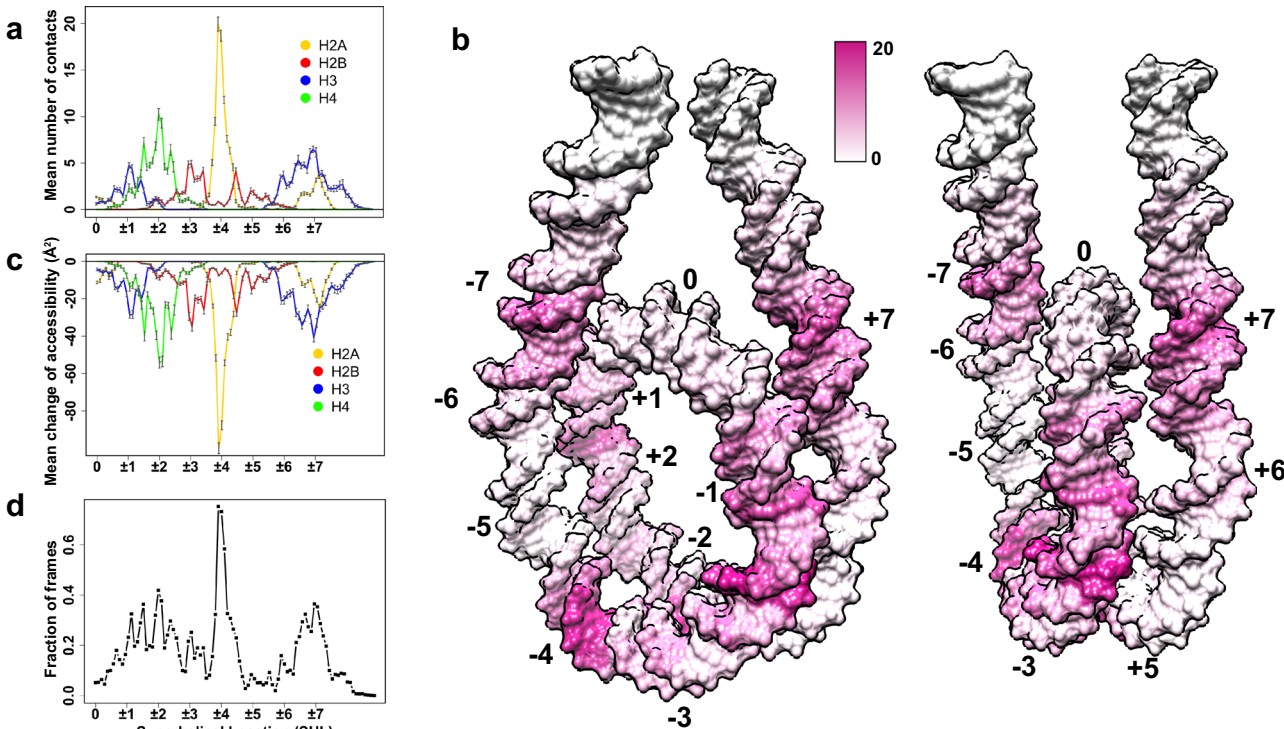

**Fig. 2 Nucleosomal and linker DNA solvent accessibility modulated by histone tail binding. a** A mean number of contacts between histone tails and nucleosomal/linker DNA averaged over all independent simulation runs for two copies ($n = 44$) plotted in the DNA coordinate frame, zero corresponds to the dyad position and superhelical locations (SHL) are shown as integers; a combined conformational ensemble from both copies of histone tails is shown. The error bars represent standard errors of the mean calculated from independent simulation runs. **b** Mean number of contacts between histone tails and DNA mapped onto the molecular surface of the nucleosomal and linker DNA. **c** Mean values of changes of DNA solvent accessibility imposed by tail binding. The error bars represent standard errors of the mean calculated from independent simulation runs for two copies ($n = 44$). **d** Percentage of frames with more than 20% SASA decrease upon tail binding per DNA base pair. Source data are provided as a Source Data file.

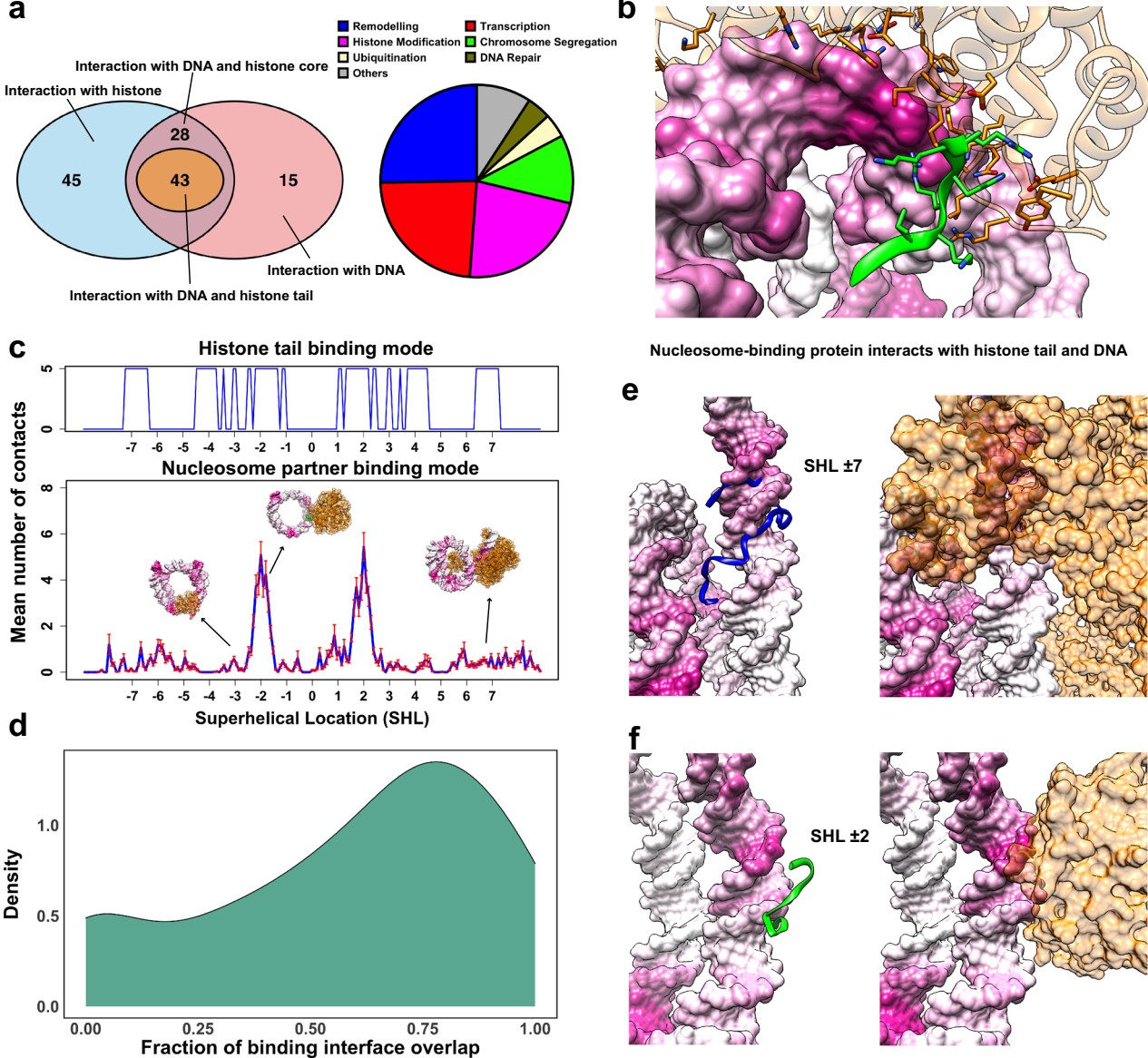

**Fig. 3 Histone tail-modulated recognition modes of nucleosomes by binding partners. a** A summary of 131 available nucleosome complex structures classified based on their binding entity and function. **b** An example of chromatin remodeler ISWI which binds to both histone H4 tail and DNA (PDB: 6IRO). Here coordinates of histone tails are taken from the PDB structure. Nucleosome-binding proteins are colored in orange and histone tails are colored using their canonical colors. **c** Analysis of the contact numbers between the nucleosomal/linker DNA and partners (averaged over all structures of nucleosome complexes) plotted in DNA coordinate frame. The error bars represent standard errors calculated from the number of contacts of different nucleosome complex structures ($n = 86$). The top track shows the presence or absence of five or more contacts between histone tails and DNA regions (indicating the histone tail preferred binding regions). **d** The fraction of interface overlap between tail–DNA and partner–DNA binding interfaces. It is calculated for each nucleosome complex structure and the distribution is smoothed using the gaussian kernel function. **e**, **f** Examples of INO80 chromatin remodeler (PDB: 6HTS) and UV-damaged DNA-binding protein (PDB: 6R8Z) targeting overlapping regions on DNA. Histone tail representative conformations are taken from simulations and superimposed onto the PDB structures. The intensity of the color of the DNA surface is scaled with the mean number of contacts between histone tails and DNA as in Fig. 2b. Source data are provided as a Source Data file.

and H4 tails, is shown in Fig. 3b. Electrostatic potential analysis shows that in this group of multivalent interactors (Fig. 4, right panel), binding partners recognize both histone tails and nucleosomal/linker DNA via two separate patches: acidic (interactions with tails) and basic (interactions with DNA). Here tails contribute positively and mediate the binding of partners to nucleosomes.

As can be seen in Fig. 3c, nucleosome-binding proteins show distinctive preferred binding regions on DNA around SHL ± 1, ±2, ±6 and ±7 and to a lesser extent on linker DNA and near the nucleosome dyad. If we compare these interfaces to the preferred

interaction modes of histone tails on DNA observed from our simulations (see the previous section), it is clear that there is a considerable interface overlap at SHL ± 1, ±2, and ±7 (Fig. 3c). Namely, dynamic histone tails and many nucleosome-binding proteins seem to target overlapping and mutually exclusive regions on nucleosomal or linker DNA. For each nucleosome complex structure where a binding partner interacts with DNA (86 complexes), we calculate a fraction of DNA interface shared between the histone tail ensemble (from MD simulations) and the nucleosome-binding proteins (from PDB structures) (Fig. 3d) and find that in 88% of them (76 complexes), interfaces are mutually

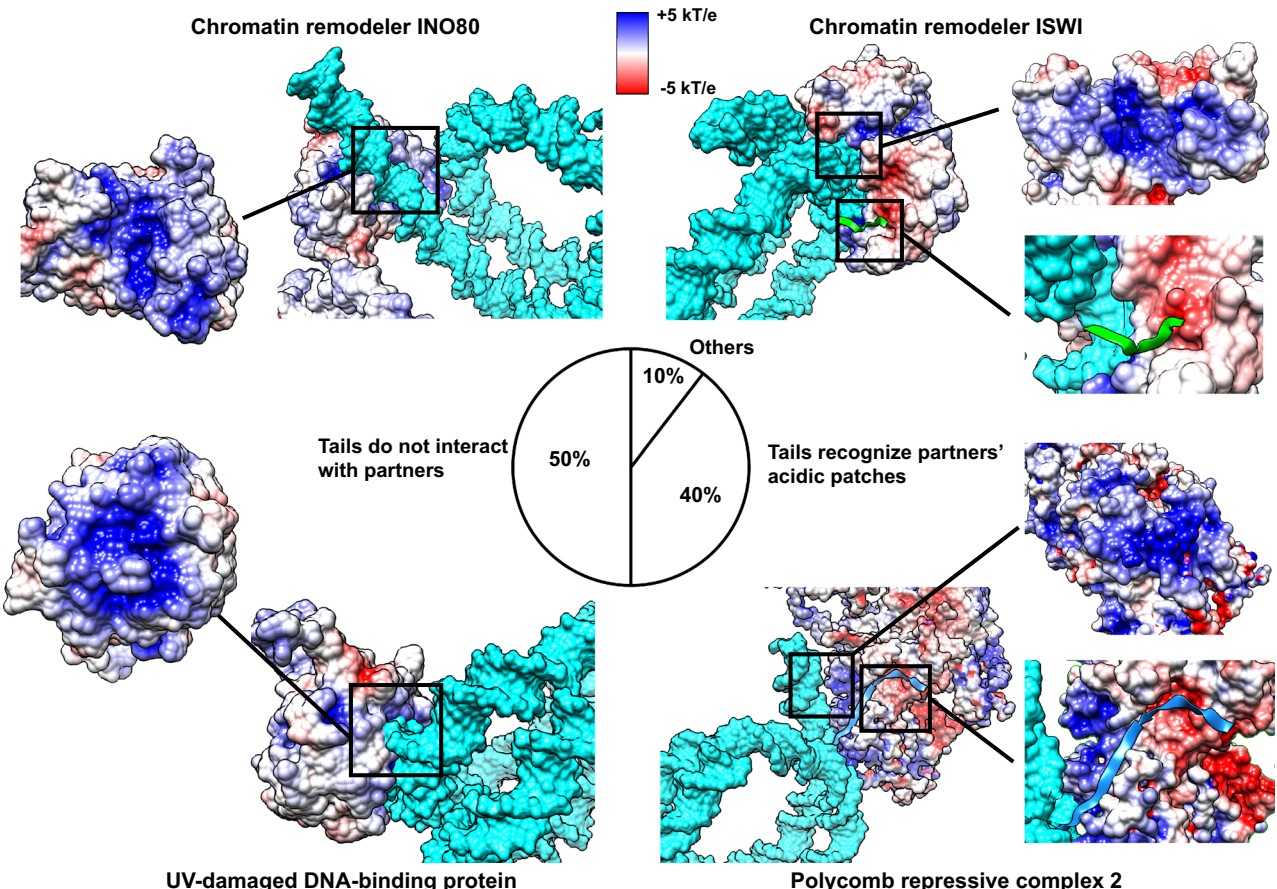

**Fig. 4 Recognition of nucleosomal DNA and histone tails by binding partners depicted via electrostatic potential analysis.** Nucleosome complex structures where proteins interact with DNA are classified based on their histone tail binding modes. Positively charged DNA-binding interfaces are highlighted for chromatin remodeler INO80 (PDB: 6HTS) and UV-damaged DNA-binding protein (PDB: 6R8Z) where partners do not interact with histone tails in structures. Another two representative examples, chromatin remodeler ISWI (PDB: 6IRO) and polycomb repressive complex 2 (PDB: 6WKR), show the DNA-binding interfaces and the partner acidic patches recognized by H3 and H4 tails. Electrostatic potentials are mapped onto the molecular surfaces of nucleosome-binding proteins. Blue and red colors indicate the positive and negative electrostatic potentials, and the intensity of the color is scaled with the surface electrostatic potential values. H3 and H4 tails are colored light blue and green. The molecular surface of nucleosomal and linker DNA is highlighted with cyan color. Core histone regions are not shown for clarity.

exclusive (at least one base pair shared on binding interfaces). Figure 3e, f show some examples, the chromatin remodeler, INO80, bound at SHL ± 7 and to the linker DNA, and the UV-damaged DNA-binding protein bound at SHL ± 2. These DNA regions can also be occupied by H3 and H4 histone tails, as evident from the tail ensemble of MD simulations shown in green and blue colors. Our electrostatic potential analysis confirms these findings showing that the DNA interfaces of these binding partners are highly positively charged, do not contain acidic patches, and are generally not very favorable for binding of like-charged tails (Fig. 4, left panel).

To elucidate how histone tails may modulate the binding of chromatin factors to nucleosomes, we consider two theoretical models. First, we estimate the equilibrium constant for binding of histone demethylase LSD1-CoREST to the nucleosome by using experimentally measured values from a recent study[31] (see Supplementary Note 2 for details). We show that histone tails' interactions with nucleosomal DNA contribute to this process, lowering the effective binding affinity. Next, we assess the scenario of the tail displacement by binding partners and report the equilibrium constant for partner–nucleosome binding being in some cases several orders of magnitude smaller than that for a partner bound to the free DNA due to the tail contribution (Supplementary Note 2). The binding of tails and partners

are controlled by both local concentrations and their binding affinities, and the effective local concentration of nucleosomes and histone tails in vivo is orders of magnitude higher than that for nucleosome-binding proteins[32].

**Histone tail post-translational modifications and mutations alter tail–DNA interactions.** Next, we try to elucidate the roles of PTMs and mutations in modulating the histone tail–DNA binding modes. Histone tails harbor different PTMs that can affect histone tail dynamics and interactions in the context of the full nucleosome. In addition, histone genes are mutated in many cancers and might represent oncogenic drivers[33]. We perform alignments of all histone protein sequences and then map nucleosome binding sites (using all collected nucleosome complex structures from PDB) and core histone cancer missense mutations from a recent histone mutation dataset onto them[33–35]. As can be seen in Supplementary Fig. 16, many cancer mutations affect the charged residues in histone tails. To further elucidate the effects of PTMs and mutations on tail–DNA interactions, we systematically compare tail-DNA interaction modes for unmodified tails and for various types of modified tails (lysine acetylation, lysine tri-methylation, serine/threonine phosphorylation, and Arg->Ala mutations) by performing 24 μs of simulations in

the context of the full nucleosome (Supplementary Table 2). Here we estimate the maximal possible effects of such modifications as these sites might not be modified at the same time in a cell.

There are two main striking observations evident from Fig. 5. First, modifications changing the effective positive charge of the residue (lysine acetylation, serine/threonine phosphorylation, and Arg → Ala mutations) significantly affect the interactions of tails with DNA (Fig. 5a), overall causing a decrease in full tail residence time $\tau_f$ (Supplementary Fig. 21). However, the amplitude of these effects depends on histone type, the position of PTM in a sequence, and modification types of the residue and other surrounding residues. The effects on H3 tail dynamic behavior are more complex: although an overall number of contacts with DNA does not change much, modifications induce the redistribution of contacts: tri-methylation of H3 introduced once at a time leads to the loss of the contacts with DNA near the dyad region and increase in the number of contacts with the DNA near the entry/exit site, SHL ± 6,7. We further assess the statistical

significance of the full tail–DNA contact number changes upon modifications (Supplementary Fig. 22). Our results show that charge changing alterations, Arg → Ala mutations, Lys acetylation, and Ser/Thr phosphorylation can lead to a statistically significant decrease in number of the full tail–DNA contacts for most tail types. However, even though certain modifications may not impact the average number of full tail contacts with DNA, these modifications may lead to a redistribution of contacts and have a significant influence on the local tail–DNA interactions, the most pronounced being the enhancement of contacts with DNA at SHL ± 2 by the H4 lysine methylation.

Second, our findings point to the crosstalk between different modified sites so that a modification in one site may lead to substantial changes of interactions with DNA in another histone site. For example, the number of contacts of H3K4 with DNA doubles when the interactions of H3R2 are suppressed through an Arg → Ala mutation. Tri-methylation of H4K5 enhances the interactions of H4R3 with DNA, whereas the interactions of

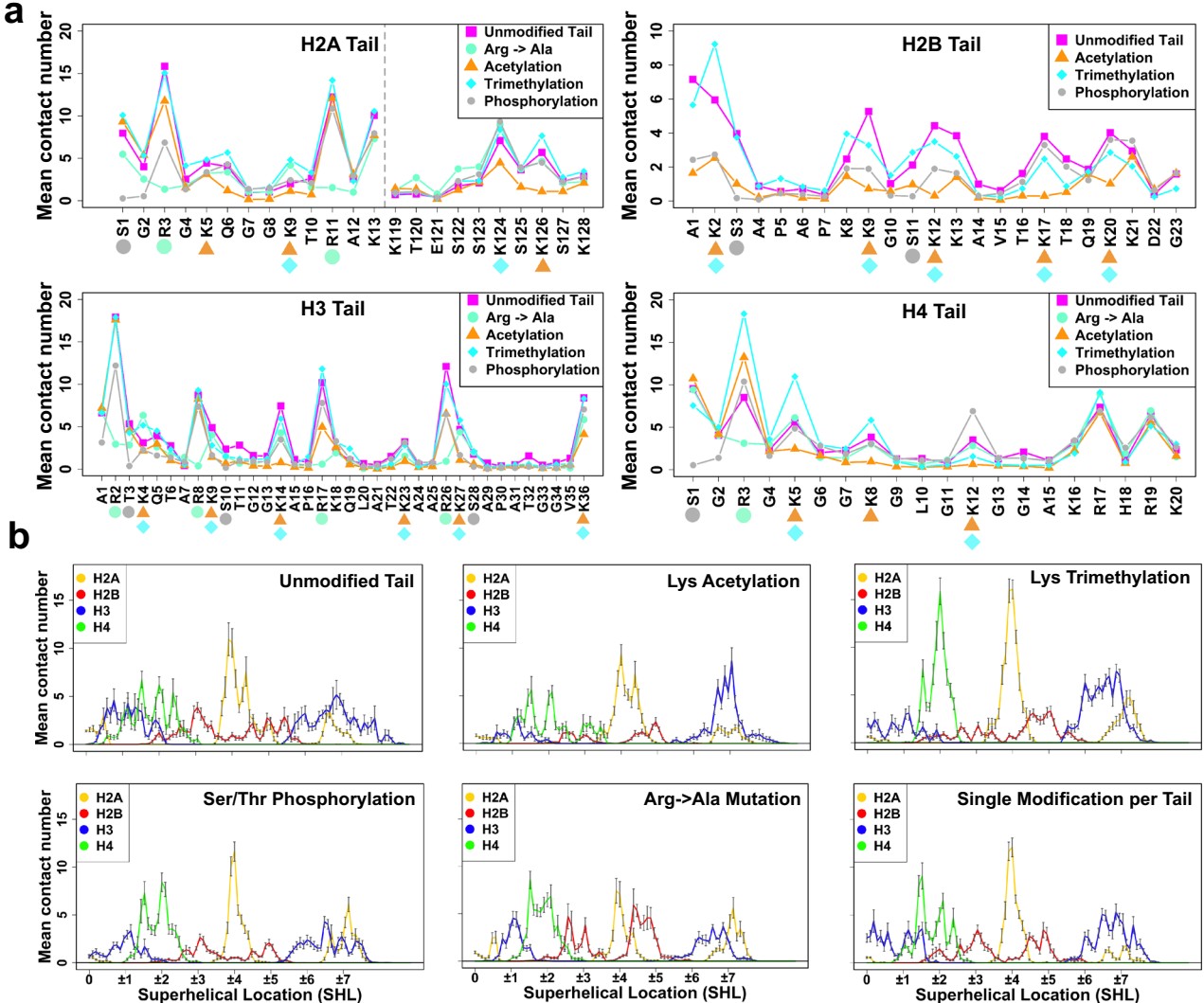

**Fig. 5 Histone tail post-translational modifications and mutations modulate histone tail–DNA interaction modes and DNA accessibility. a** Mean number of histone tail–DNA contacts for different types of modifications per residue. The result of simulations of model D with and without corresponding modifications are shown in different colors and modifications are shown by a symbol next to the residue. **b** Mean numbers of nucleosomal and linker DNA contacts with histone tails for different types of modifications per DNA base pair. For each type of modification, the reported values are averaged, and the error bars represent the standard errors of the mean calculated from independent simulation runs for two copies (*n* = 10). The locations and types of modifications and mutations are listed in Supplementary Table 2. The mean number of contacts of unmodified tails was calculated using the first 1600 ns frames from the trajectory of the simulation of Model D with the AMBER package (Supplementary Table 1). Source data are provided as a Source Data file.

H3R26 with DNA are suppressed by phosphorylation of H3S28. Interestingly, we also observe the crosstalk between different tail types where modifications in one tail affect the tail–DNA interactions in another tail: a suppression of H2A N-terminal tail interactions through Arg → Ala mutations can lead to a significant increase in DNA binding of H2B tails (which do not carry Arg → Ala mutations) which occupy similar regions on DNA (Fig. 5b).

## Discussion

Nucleosomes are elementary building blocks of chromatin and, at the same time, may act as signaling hubs by integrating different chromatin-related pathways[27] and directly participating in the regulation of many epigenetic processes pertaining to the access of chromatin factors to DNA and histones[27]. It has long been debated that the DNA solvent accessibility and mutability can be modulated for those regions which are packed in nucleosomes[6]. According to the commonly used static model, the DNA accessibility follows the 10 base pair periodicity patterns of rotational positioning of nucleosomal DNA[36]. However, we have shown that there is another important layer in this mechanism, which stems from the histone tail dynamics. Even though histone tails extensively condense on the DNA, comprehensive simulations performed in this study allow us to observe many histone tail binding and unbinding events. Namely, we demonstrate that the tails undergo rapid transitions between bound and unbound states, and the kinetics of these processes depend on the histone type. The interactions between tails and DNA are transient, and switching between tail conformations occurs on the time scale from tens and hundreds of nanoseconds to several microseconds in the form of jittery motions, with the H2A C-terminal tail having the shortest residence time on DNA and H3 tail having the longest residence time. The emerging body of experimental evidence has pointed to the high level of conformational dynamics of histone tails, with the dynamic conformational transitions on the order of sub-microseconds[9,37,38]. This is consistent with our observed highly dynamic tail behavior from simulations. In addition, a recent FRET study shows that H3 tails have multiple interaction modes with the nucleosomal or linker DNA, with conformational transitions from compact to extended states taking place on micro- to millisecond timescales[39]. Our results also show that compared to other tails, the H3 tail has the longest residence time on DNA, on the order of microseconds or longer. The total binding free energy of histone tails is generally proportional to the tail length and effective positive charges, and the longest and arginine-rich H3 tail has the highest DNA-binding affinity. Our results show that anchoring of arginine, followed by lysine and serine, is critical in determining the tail's residence time. A decrease of the effective charges or increase of hydrophobicity in histone tails dramatically suppresses the tail–DNA interactions and decreases the tail residence time on DNA.

The interactions of histone tails with the DNA molecule within the same nucleosome affect the nucleosomal and linker DNA accessibility—even though the interactions of individual tails with DNA are transient, DNA regions SHL ± 1, ±2, ±4, and ±7 can be partially or substantially occluded from the solvent by different types of histone tails. Histone tail interactions with DNA may modulate the accessibility of both DNA and histone tails themselves to other binding biomolecules[11,40]. Indeed, a recent study has shown that the PHD fingers of CHD4 bind up to 10-fold tighter to histone tail peptides compared to binding to the tails in the context of nucleosome[18]. A recent large-scale experimental study also demonstrates that many chromatin factors show enhanced binding to tailless nucleosomes compared to the full

nucleosome, resulting from the increased solvent accessibility of DNA[41]. Our estimates of the standard state binding free energy of the histone tail binding to DNA based on conformational sampling are on the order of several kcal/mol for H3 and H4 tails with the strongest binding exhibiting for H3 tail (Supplementary Table 4). Recent quantitative experimental studies reported standard binding free energies of partners to nucleosomes and DNA being in the range of 8–10 and 3–15 kcal/mol, respectively (Supplementary Table 9, Supplementary Fig. 23)[42,43]. It should be mentioned that the probability of binding of tails and partners is controlled by both local concentrations and their binding affinities, and the local concentrations of histone tails and nucleosomes are orders of magnitude higher than the concentration of the nucleosome-binding partners.

Based on the analysis of dynamic MD ensemble of histone tail conformations and nucleosome experimental structural complexes, our results further indicate that nucleosome-binding proteins and histone tails may target overlapping and mutually exclusive regions on nucleosomal or linker DNA around SHL ± 1, ±2, and ±7. This trend is observed for 76 studied nucleosome-binding partners and points to a potential competitive binding mechanism: nucleosome-binding proteins compete with DNA if they recognize tails and compete with histone tails for binding to DNA (Fig. 6). The competition between chromatin factors has been previously recognized as a major determinant of various chromatin states[44]. At the same time, our analysis identifies 43 structures of nucleosome-binding proteins interacting with both histone tails (via H3, H4, and H2A-C terminal tails) and nucleosomal or linker DNA (Supplementary Table 8) and in these complexes, histone tails do not have direct contacts with DNA. Such recognition patterns could be explained by multivalent binding and/or by a recently proposed tail displacement model (Fig. 6)[31,45]. According to the tail displacement model, interactions of DNA-binding domains (DBD) of a nucleosome-binding protein with the nucleosome can displace histone tails from their DNA preferred binding modes. It makes tails more accessible for recognition by reader domains (Fig. 6). The displacement of histone tails, in turn, can be facilitated by the competitive binding between histone tails and DBDs if they both recognize the same regions on DNA. This could accelerate the unbinding of tails from DNA and enhance the recognition of tails by the reader domains. This is supported by recent studies showing the displacement of H1 C-terminal tail and H3 tails induced by binding of HGMN protein to nucleosomes[46] and by another study showing the competitive binding between chromatin remodeler ISWI and H3 tails[47].

Histone tail post-translational modifications can be responsible for the regulation of tail–DNA interactions through the alteration of histone tail binding modes (Fig. 6). As we demonstrate, charge-altering modifications and mutations in histone tail residues overall may suppress tail–DNA interactions and enhance histone tail dynamics and DNA accessibility. Consequently, this mechanism can boost the interactions between nucleosomes and nucleosome-binding proteins, which specifically recognize certain histone tail sites and/or nucleosomal/linker DNA. Consistent with these observations, phosphorylation, and acetylation of H3 tails were found in recent studies to weaken H3 tail–linker DNA interactions to stimulate the H3 tail dynamics[11,15]. We show that histone modifications may have local or long-distance effects, and modification in one site can influence the dynamics and histone–DNA interactions in another site. As an example, interactions of arginine residues with DNA can be modulated by trimethylation of lysine located up to several residues apart in sequence.

Beyond the intra-nucleosomal interactions, tail–DNA interactions have long been indicated to play critical roles in inter-

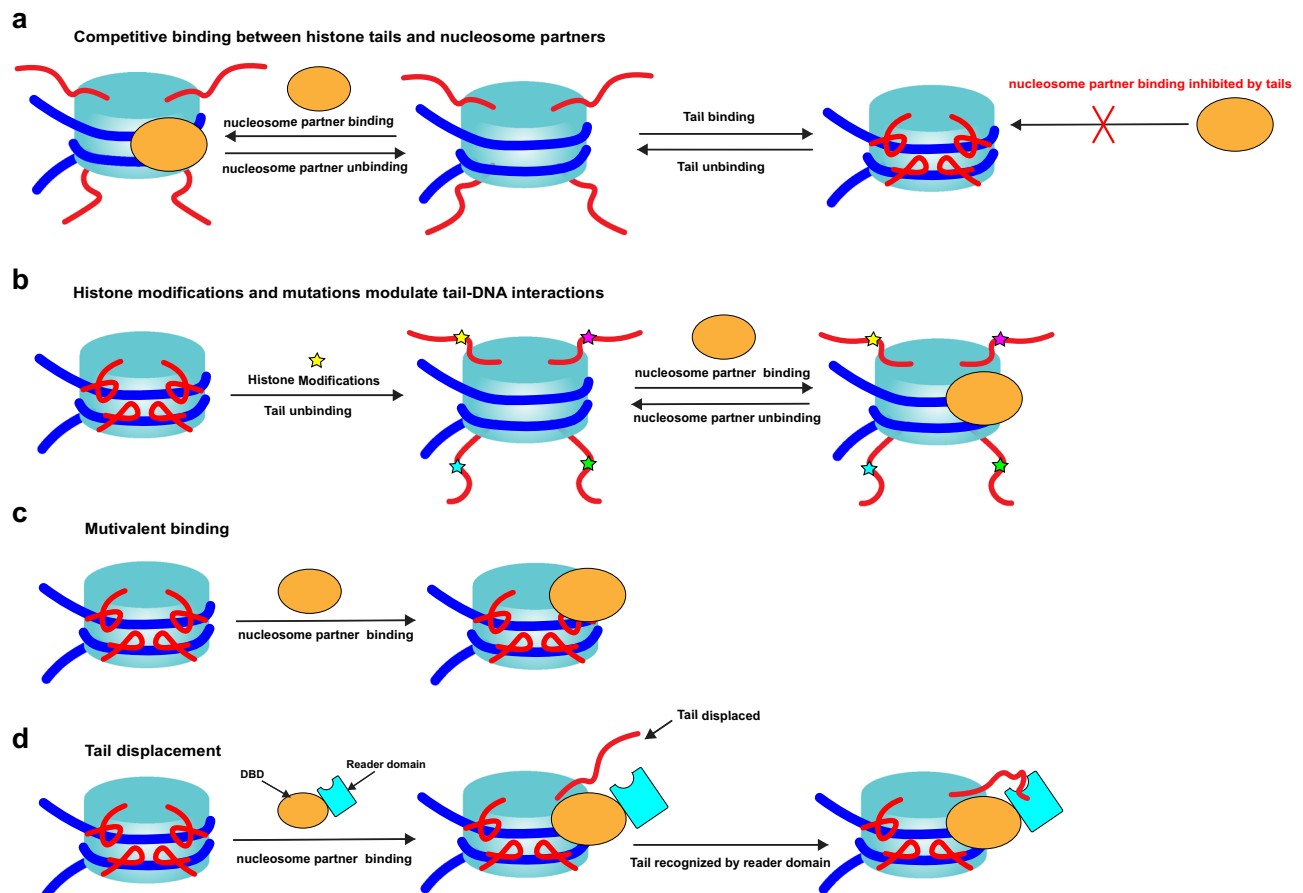

**Fig. 6 A generalized model explaining how tails and their modifications can modulate nucleosomes' interactions with nucleosome-binding proteins.**
**a** Histone tails' interactions with DNA may in some cases modulate the accessibility of DNA to binding partners; nucleosome-binding proteins compete with histone tails for binding to DNA. **b** Post-translational modifications and mutations in histone tails can suppress tail–DNA interactions, enhance histone tail dynamics, and regulate the binding of proteins to nucleosomes. **c** Some nucleosome-binding proteins may exhibit multivalent binding modes and recognize both histones and DNA. **d** DNA-binding domains (DBDs) of nucleosome-binding proteins compete with histone tails if they recognize DNA and reader domains compete with DNA if they recognize tails; interactions of DBD with nucleosome can displace histone tails from their DNA preferred binding modes and increase their accessibility for recognition by reader domains.

nucleosome interactions[37,48]. It has been shown that H3 and H4 tail–DNA interactions are important for compaction and oligomerization of nucleosome arrays[49]. Previous NMR studies have further shown that histone tails have a high degree of conformational flexibility even in a highly compact chromatin state[10], similar to the tail conformations in a single nucleosome. As in the intra-nucleosome interactions, charge-altering modifications or mutations can suppress the tail–DNA interactions and affect the inter-nucleosome interactions and the higher-order chromatin structures. Along these lines recent studies have suggested that H4 tail acetylation can not only suppress the tail–DNA interactions but also lead to the compaction of tail conformations, reducing interactions of tails with neighboring nucleosomes[50,51].

We argue here that histone tails are crucial elements in coordinating the transient binding and recognition of different chromatin factors to nucleosomes and thereby contribute to the regulation of epigenetic processes in time and space. Their disordered dynamic nature is a prerequisite for allowing histone tails to bind to different partners via the same interface with high or low affinity and high specificity. Similar to well-documented cases of the disorder-mediated control of the exposure of protein–protein interfaces, here we argue that an analogous mechanism can pertain to protein–DNA interface exposure at the level of the nucleosome and show that modulating DNA access through histone tails might represent a rather general

mechanism. The quantitative characterization of these dynamic processes is very challenging, and data is still largely lacking. The future focus on the development of experimental and computational techniques elucidating the spatial and temporal hierarchy of dynamic chromatin processes may close this gap in our understanding.

## Methods

**Construction of full nucleosome models with the native DNA sequence.** There have been very few native genomic DNA sequences used in experimental and computational studies of nucleosomes[52]. Here, we constructed a structural model of a nucleosome with the DNA sequence from a well-known oncogene, *KRAS*. In order to do this, we first identified the precise translational positioning of DNA with respect to the histone octamer. To determine the dyad position of the nucleosome, we applied a previously developed nucleosome mapping protocol to Micrococcal nuclease (MNase) experimental data using the hg19 human genome assembly[53]. Fragments of 147 bp lengths of high-coverage MNase-seq reads were used (available in the GEO under accession number GSE36979), and the dyad positions were determined as middle points of these fragments[54]. The mid-fragment counts were smoothed out using a 15-bp tri-weight kernel function to get the kernel-smoothed dyad count. Then, the dyad positions with local maximum values of the smoothed counts were obtained using bwtool[55] and the dyad with the highest number of counts within a 30 bp interval was selected as the representative dyad. Next, we identified a well-positioned nucleosome as the first nucleosome positioned downstream of the transcription start site (the +1 nucleosome of the *KRAS* gene). To create a structural model of the full nucleosome with the DNA linkers, we used a high-resolution X-ray structure of a nucleosome core particle (NCP) formed by *Xenopus laevis* canonical core histones and human α-satellite

sequence (PDB:1KX5)[56]. Then we linearly extended DNA from both ends by adding 20 bp linker segments using the NAB software (one of the H3 tails was slightly rotated to avoid steric clashes with the linker DNA by setting $\psi$ angle of Lys36 to $-35°$)[12,57]. The native DNA sequence was selected from the human genomic region centered around the $KRAS + 1$ nucleosome dyad and flanked by the 93 bp segments on each side (Supplementary Table 10). Finally, we embedded the native DNA sequence onto the structural nucleosome model using the 3DNA program[58].

There are several structures in PDB which contain coordinates of partially resolved histone tails, which can be used in the in silico studies of nucleosomes. However, histone tails are intrinsically disordered, and their conformational ensemble covers a wide spectrum of possible configurations. We constructed several nucleosome models with different initial tail configurations and used them for simulations. First, we explored the existing high-resolution NCP structures (with a resolution higher than 3 Å) with the full or partial histone tail atomic coordinates in PDB[59], out of which two structures (PDB:1AOI and PDB:1EQZ) were selected based on their high resolution and partially solved histone tails. H3 and H4 tail coordinates were taken from 1AOI, and one H3 tail and two H2B tails coordinated from 1EQZ, while the conformations of other tails were taken from structure 1KX5. In those cases where templates did not contain all residue coordinates at the end of histone tails, missing residue coordinates were modeled by linearly extending existing tail conformations (dihedral angles for each residue were $\Phi$ angle $= -60°$ and $\Psi$ angle $= 30°$). As a result, two models (Models A and B) were built.

Furthermore, we constructed two additional models by linearly extending histone tails from the histone core into the solvent. Namely, we clipped all tails from the original 1KX5 structure at sites H3K37, H4K16, H2A A12-K118, and H2BK24 following histone tail definition from[12] and then tails were linearly reconstructed using the building structure plugin in Chimera[60] (dihedral angles used for each residues $\Phi = -60°$ and $\Psi = 30°$). In one initial model (Model C), tails were extended from the histone core following the backbone orientation of the last two residues at the truncated sites. We also built another initial model where histone tails were extended into the solvent symmetrically oriented with respect to the dyad axis (Model D). The Modeler software was used to remove steric clashes in tail residues surrounding the truncated sites[61]. Overall, we constructed four models with different initial tail conformations for simulations (Supplementary Table 1 and Supplementary Fig. 1).

**Choice of force fields, water models, and ion parameters**. An appropriate choice of the force field, water model, and ion parameters is required to simulate highly charged large macromolecular systems such as nucleosome, to model protein–DNA interactions, conformations of disordered histone tails, and nucleosome interactions with the solvent and ions.

Here, we use recently developed protein and DNA force fields: AMBER ff14SB force field for protein and OL15 force field for DNA[62–65]. We use an optimal point charge (OPC) water model with the 12-6 HFE parameter set for monovalent ions. OPC is a 4-point rigid water model, which has been shown to reproduce comprehensive sets of water liquid bulk properties and delivers noticeable accuracy improvement in simulations of DNA and RNA, thermodynamics of ligand binding, small molecule hydration, and intrinsically disordered proteins[66–68]. The OPC water model, together with the AMBER force field, offered remarkable improvements over the TIP3P water model in the modeling of the conformational ensembles of IDPs[69]. Most recently, the OPC water model was applied in simulations of chromatosomes[47]. For preparatory short simulations, we also used protocols with the TIP3P water model, CHARMM force field, and Beglov and Roux ion parameters[70], although did not report the results for reasons outlined in the "Results" section (Supplementary Note 1).

For four constructed nucleosome models (Model A–D), we performed simulations using the AMBER and CHARMM force fields. For the AMBER simulations with the OPC water model, for each nucleosome model, we performed five independent runs with different seeds, four runs had 800 ns simulation time, and one run reached 4000–5000 ns for the purpose of observing phenomena on a longer time scale. For model D (nucleosome model with the symmetrically extended tails), we performed two 5000 ns simulation runs using GROMACS with the OPC water model and AMBER force field. In parallel, we performed three 100 ns simulations for each nucleosome model using the CHARMM force field and the TIP3P water model. A summary of all simulation runs for histone tail sampling is provided in Supplementary Table 1.

**Simulation protocols**. The MD simulations using the AMBER force field and OPC water model were prepared and performed with the Amber18 package[71] and GROMACS version 2019.3[72]. MD simulations using the Amber18 package (20 simulations runs in total, 4–5 μs each) were performed as following (Supplementary Table 1). Nucleosome structures were solvated with 0.15 M NaCl in a cubic water box with at least 20 Å from the protein to the edge of the water box (detailed information is provided in Supplementary Table 1). Systems were maintained at $T = 310$ K using the Langevin dynamics with the integration step of 2 fs and collision frequency $\gamma = 1$ ps$^{-1}$. The Berendsen barostat was used for constant pressure simulation at 1 atm. SHAKE bond length constraints were applied for bonds involving hydrogens. The cut-off distance for non-bonded interaction

calculations was 10 Å. Particle mesh Ewald (PME) method with a spacing of 1 Å and real space cut-off of 12 Å was applied for the electrostatic calculations. Periodic boundary conditions were used, and the trajectories were saved every 20 ps. All systems were first subjected to 10,000 steepest descent minimizations and then for another 10,000 conjugate gradient minimizations. After minimization, systems were gradually heated from 100 to 310 K in the NVT ensemble and then switched to the NPT ensemble for 500 ps equilibrations before production runs.

Two simulation runs using the GROMACS package were performed as following (Supplementary Table 1). A cut-off of 10 Å was applied to short-range nonbonded interactions, and the PME method was used in calculating long-range electrostatic interactions. Long-range dispersion corrections for energy and pressure were applied for long-range Van der Waals interactions. Covalent bonds involving hydrogens were constrained to their equilibrium lengths using the LINCS algorithm. The solvated systems were first energy minimized using steepest descent minimization for 10,000 steps, gradually heated to 310 K over the course of 800 ps using restraints, and then equilibrated for a period of 1 ns. After that, the production runs were carried out in the NPT ensemble up to 5 μs, with the temperature maintained at 310 K using the modified Berendsen thermostat (velocity-rescaling) and the pressure maintained at 1 atm using the Parrinello–Rahman barostat.

**Simulations of nucleosomes with mutated and post-translationally modified histone tails**. To elucidate the effects of mutations and histone modifications on tail–DNA interactions, we performed multiple sets of simulations, including lysine acetylation, lysine trimethylation, serine/threonine phosphorylation, and Arg → Ala substitutions introduced at the same time or at one residue at a time (Supplementary Table 2). We used the constructed nucleosome structure Model D (histone tails were extended into the solvent symmetrically oriented with respect to the dyad axis) for these simulations. AMBER force field and OPC water model were applied using protocols described above. Mutations and PTMs were introduced to nucleosome structures with LEaP in the AMBER package[71], and locations and force field parameters of PTMs were taken from previous studies[73,74]. For each set of simulations, we performed five independent runs with different random seeds, of which four runs had 800 ns simulation time and one run of 1600 ns.

**Trajectory analysis**. Trajectories were visualized and analyzed using a set of TCL and Python scripts that utilized the capabilities of VMD[75], 3DNA[58], and AMBER Tools[71]. The trajectory frames were superimposed onto the initial models by minimizing RMSD values of $C^\alpha$ atoms in histone cores (Supplementary Table 3). In the analysis of histone tail–DNA interactions, tail–DNA atomic contacts were calculated for trajectory frames of every 1 ns. The first 200 ns frames of each 4000–5000 ns run and 50 ns frames of each 800 ns run were disregarded as an initial conformational equilibration period. The contacts of atoms between histone and DNA were defined between two non-hydrogen atoms located within 4 Å. For each DNA base pair, we calculated the mean number of bound histone tail heavy atoms averaged over frames. Then, we defined the histone tail preferred binding regions on DNA as those DNA base pairs that had more than five contacts on average with histone tails.

The residence time of histone tails was defined as the time during which tails remained bound to DNA in the simulations. Two types of residence time were calculated: individual residue residence time ($\tau_r$) and full tail residence time ($\tau_f$). Since unbinding of entire histone tails occurs on a relatively long timescale, we only used the trajectories from the long runs (4000–5000 ns) for calculating the full tail residence time. Here, a bound state for an individual residue was defined if at least one heavy atom of a residue had contact with DNA. An unbound state for the full tail was defined if no more than a certain fraction of histone tail residues maintained contact with the DNA molecule (different values of this threshold were tested; see Supplementary Materials). Since full histone tails undergo very rapid fluctuations before retaining stable binding with DNA during the simulations, we ignore $\tau_f$ of shorter than 10 ns. DNA solvent accessibility surface area (SASA) was calculated using VMD[75] with a probe distance of 1.4 Å for every 5 ns frames. The nucleosomal and linker DNA SASA change upon histone tail binding was calculated as the difference between the SASA of DNA with tails bound to it and without tails. The percentage of accessibility change for a DNA base pair is defined as a difference between SASA of nucleosomal/linker DNA with and without bound tails divided by the total SASA.

The binding free energy between histone tails and DNA was calculated using the molecular mechanics generalized Born surface area (MM/GBSA) method implemented in the Amber18 package. We performed calculations for every 1 ns frame (ignoring the first 50 ns in the 800 ns trajectories and 200 ns in 4000–5000 ns trajectories), and residue-wise decomposition was applied to derive the binding energy per tail residue. Each copy of a tail within a simulation was considered as a separate observation of the tail ensemble. Thereby there were two conformations per frame per histone type. In all calculations, the standard error (SE) of the mean from independent simulation runs for two copies (22 runs in total) were estimated.

**Analysis of experimental structures of nucleosomal complexes**. We extracted all nucleosome complex structures from PDB[30] for our analysis of nucleosome-binding proteins and then removed 20 structures that did not contain the complete

histone octamer or had extensive DNA unwrapping or sliding along the octamer (structures where proteins interacted with the linker DNA, were kept in our analysis). In total, we analyzed 131 nucleosome complex structures. The interaction between a nucleosome and a binding partner was defined if histone proteins and/or nucleosomal and/or linker DNA had at least one non-hydrogen atom within 4 Å of nucleosome-binding proteins. Functional classifications of nucleosome binding proteins were performed using the general protein function annotations from UniProt[76]. To quantitively characterize the degree of DNA interfacial overlap between DNA–histone tails and DNA-nucleosome binding proteins, we calculated the fraction of interface overlap as a number of DNA base pairs found on both DNA–tail preferred binding regions (from MD simulations) and DNA-partner binding interfaces (from PDB experimental structures) divided by the number of DNA base pairs making contacts with nucleosome-binding proteins in a PDB structure. The histone tail preferred binding regions on DNA are defined as those DNA base pairs that have more than five contacts on average with histone tails in MD simulations.

**Electrostatic potential calculation**. The electrostatic potential of nucleosome binding proteins was calculated using the Delphi program[77]. The dielectric constant for protein and solvent was set to 2 and 80, respectively, and the salt concentration was 0.15 M. The percentage filling of the box was 70 with a scale of 2 grid/Å, and the water probe radius was 1.4 Å. The calculated potential map was saved in CUBE format and was further visualized using UCSF Chimera[60].

**Reporting summary**. Further information on research design is available in the Nature Research Reporting Summary linked to this article.

## Data availability
The data that support this study are available from the corresponding author upon reasonable request. Source data are provided with this paper and available from GitHub at https://github.com/Panchenko-Lab/Supplementary-data-for-Peng-et-al-2021. Molecular dynamics simulation trajectories generated in this study are archived via Zenodo at https://doi.org/10.5281/zenodo.4771269. Fragments of 147 bp lengths of high-coverage MNase-seq reads used in this study are available in the GEO under the accession number GSE36979. Nucleosome structures used in this study are available in Protein Data Bank (https://www.rcsb.org). PDB IDs of analyzed nucleosome complex structures are provided in Supplementary Table 11. Source data are provided with this paper.

## Code availability
Computer code is available at https://github.com/Panchenko-Lab/Supplementary-data-for-Peng-et-al-2021.

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

## Acknowledgements

Y.P. and D.L. were supported by the Intramural Research Program of the National Library of Medicine at the U.S. National Institutes of Health. A.R.P. was, in part, supported by the Intramural Research Program of the National Library of Medicine at the U.S. National Institutes of Health. S.L. and A.R.P. were supported by the Department of Pathology and Molecular Medicine, Queen's University, Canada. A.O. was supported by the funding from NIH R21GM134404. A.R.P. is the recipient of a Senior Canada Research Chair in Computational Biology and Biophysics and a Senior Investigator Award from the Ontario Institute of Cancer Research, Canada. This study utilized the high-performance computational resources from the Biowulf cluster at the National Institutes of Health (https://hpc.nih.gov/systems/) and Compute Canada (https://docs.computecanada.ca).

## Author contributions

Y.P. performed simulations with the help of S.L. Y.P. performed all analyses. D.L. and A.O. contributed to the study design and paper editing. D.L. supervised and contributed to funding acquisition. A.R.P. and Y.P. wrote the paper with the support from all authors. A.R.P. supervised and conceived the project and contributed to funding acquisition.

## Competing interests

The authors declare no competing interests.
