## [Peer Review File · Nature Communications]

REVIEWER COMMENTS

Reviewer #1 (Remarks to the Author):

In this manuscript, the authors carried out extensive atomistic simulations to study the interactions between histone tails and nucleosomal DNA. They observed rapid interconversion of bound and unbound states and estimated the residence time and binding free energy. The authors further analyzed structural data of nucleosome protein complexes and found that the binding sites overlap with regions occupied by histone tails. Post-translation modifications can weaken tail DNA interactions and enhance histone tail dynamics. While these simulations themselves are interesting, I have reservations about their convergence and interpretation.

First, while 26 microseconds may sound impressive and are indeed heroic efforts for atomistic simulations, they do not necessarily support statistical convergence. There are encouraging signs, including the two-fold symmetry shown in Figure SM5 and the correlation between binding free energy from simulations and MM/GBSA calculations. However, what concerns me is that the two water models give rise to significant differences in binding and unbinding kinetics. While the authors argue that the OPC model is expected to outperform TIP3P, most references appear to be on water and protein simulations. Is there evidence that supports the OPC model's performance for protein-DNA interactions?

The authors explored different ways to build initial structures for histone tails to avoid biases in their prediction of the binding sites. It would be useful to analyze the difference among results from these initial configurations. Do they more or less converge to the same binding sites, or are there significant differences among them? These analyses could provide more insight into the statistical convergence.

Second, given that the histone tails cover almost the entire nucleosomal DNA (Figure SM5), it's perhaps not too surprising that there is some overlap between tail and chromatin regulator. The implications of this observation are nontrivial. I am not convinced that histone tails serve as roadblocks that must be displaced for protein binding. In particular, histone tails are flexible and adopt rather dynamic binding sites. They could adjust their conformation to accommodate the additional proteins or seek new binding sites. The delta G values in Figure SM4 have no units, but if I interpret that as kcal/mol, then 3kcal/mol for H4 is not a significant number, especially considering that it's for unbinding of the entire tail. These estimations also ignored the protein-tail interactions, which could contribute positively to the binding of nucleosomal proteins.

Mino point:

- 1) Numerous prior studies have observed a change in histone tail secondary structure upon DNA binding. It would be useful if the authors could comment on the corresponding changes in their simulations.
- 2) The font size used in many figures is too small and not legible.
- 3) I would be more reserved with the sentence "Overall, it shows a remarkable linear association with $R^2 = 0.9$, pointing to a reasonable conformational sampling of histone tails' binding performed in our study (Figure 1d)". The correlation requires fine-tuning the cutoff parameter, and the values for H3 cannot be estimated reliably.
- 4) SHL was never clearly defined.

Reviewer #2 (Remarks to the Author):

This manuscript describes a set of molecular dynamics simulations that investigated the spatial distribution of full-length histone tails around a nucleosome particle. Probing such distributions using the all-atom MD method has been difficult in the past because of the insufficient simulation time scale and because of the force field artifacts that favored overly compact conformations for intrinsically disordered proteins. The authors have overcome the latter problem by using a custom water model developed by the Onufriev lab. Indeed, instead of seeing a permanent binding of a histone tail to DNA, which is a typical outcome of a simulation performed using either standard AMBER or standard CHARMM force field, the authors observed a multitude of binding and rebinding events, enough to draw statistically significant conclusions on where and for how long the histone tails bind. Armed with that information, the authors find a significant overlap between the DNA regions where the histone tails bind and the DNA regions occupied by DNA binding proteins in the available crystallographic structures. Interestingly, chemical modifications of histone tail residues were found to alter the histone tail binding in a non-trivial manner. The authors conclude that histone tail binding to DNA can regulate accessibility of the nucleosomal DNA to chromatin remodeling factor and other DNA binding proteins.

This is an interesting study that tackles head on an outstanding question in chromatin biophysics. The simulations were carefully designed and carried out in replicate to improve statistical significance of the results. The manuscript text is clearly written, but the figures require improvement. The authors are asked to revise the manuscript to address the following.

Perhaps the only major criticism of the work is its indirect conjecture on histone tail binding being able to outcompete or reduce protein binding to nucleosomal DNA. The histone displacement model provides a believable route to how protein binding to DNA can proceed despite the DNA being occupied by histone tails. Histone tail might also have a favorable affinity to DNA binding proteins, so it is not at all clear to which degree DNA occlusion translates into modulation of DNA binding affinity. Ideally, the protein binding scenario should have been investigated through free-energy calculations. This reviewer, however, realizes the challenges involved in carrying out such multi-dimensional calculations. A simple resolution could be a theoretical model that, taking the affinity of a DNA binding protein to the cognate DNA fragment and to the histone tails as input, predicts the effective binding constant for the protein. The results of such model could be presented in a new figure, giving the readers an idea about the range of conditions where DNA occlusion by histone tails becomes important.

The 250 ns simulation time scale is a bit short for 2021. Any chance all simulations could be run for 2.5 microsecond? If not, consider extending to what is technically feasible while working on the revisions.

The authors accumulated significant amount of data on tail binding to DNA. Can the authors say anything general on how the binding depends on the tail length, charge and/or hydrophobicity?

Previous simulations and experiments found AT-content to correlate with the affinity of poly-lysine peptides to DNA [Nucleic Acids Research 46: 9401]. Do the authors see any sequence/AT content preference for histone tail binding?

Please clarify the initial conformation used for the simulations of the modified tails systems.

The effect of modifications on tail binding is striking. What is lacking is some kind of statistical measure of its significance. Ideally, these simulations should be run longer.

Page 12, first sentence. Panel f is not referenced before Panel b.

Page 12, it would be good to specify typical Delta G values for protein binding to DNA here and compare them to tail binding values.

Lines 403-405: This sentence implies that histone tails facilitate binding of chromatin factor, in contrast to the next sentence. Maybe, to emphasize that fact better, start the next sentence with “In contrast”?

Line 413 starts with “Our second prediction”. What was the first one? The previous paragraph does not explicitly describe one.

Line 426: Please replace “them” with either “tails” or “modes”, depending on the intended meaning.

Figure 1 and most of the figures. The axis labels are way too small, almost microscopic. The figures appear pixelated (a pdf to MS word conversion, likely), difficult to extract quantitative information.

Figure 1d: The MM/GBSA free energies are way too large (by magnitude) to be realistic. The biotin-streptavidin binding energy of -18kcal/mol guarantees almost irreversible binding. A binding energy of -50kcal/mol signifies a permanent attachment. Maybe remove MM/GBSA results?

Figure 2a, c and d: Define SHL

Figure 2d: The Y axis should probably be “fraction” not “percentage”

Figure 3b is confusing. The top panel does not have any axes or units. The title above panels does not help.

Figure 3c: The Y axis (density) should have some kind of units.

Figure 3 caption. In b, start the caption with “Crystallographic analysis of ... “ or something similar.

Figure 3 caption. In c, the caption mentions smoothing with a kernel, what were the parameters of the kernel?

Figure 4: Axes are missing in panel a

Reviewer #3 (Remarks to the Author):

Binding or regulatory to nucleosomes is modulated by dynamic histone tails

By Pachenko et al.

This comprehensive simulation work on effects of PTM on NCP histone tail modifications is significant. My only concern is that the length of simulation for PTM and oncogenic mutations (2000 ns) may not allow for the relaxation of the tail positions. Computational results for PTM effects on binding affinity of tail are reported and compared to experimentally reported results. This article should be considered for publication in Nature Communications after some major concerns are addressed:

1. Binding and unbinding behavior of the tails reported here is entirely force field dependent. This is quite interesting. What validation is there that the Amber OL15 force fields more accurately reproduce timescales of experimental binding/unbinding?
2. More discussion concerning significance of PTM on larger scale chromatin structure and phase behavior should be reported. All effects of PTM are discussed on single nucleosome level. How would this affect nucleosome-nucleosome interactions, etc? How would PTM affect the dynamics?
3. Minor Concern: Figure SM4 Unbinding is spelled incorrectly. Figure 1 contains too many panels and the captions are exceedingly hard to read. This figure should be split up and a portion can be moved to the SI.

RESPONSE to REVIEWERS' COMMENTS

We thank all reviewers for their thoughtful and constructive comments and suggestions, which helped us to strengthen our paper significantly. Our detailed responses to the referees' comments are itemized below.

Reviewer #1:

1. In this manuscript, the authors carried out extensive atomistic simulations to study the interactions between histone tails and nucleosomal DNA. They observed rapid interconversion of bound and unbound states and estimated the residence time and binding free energy. The authors further analyzed structural data of nucleosome protein complexes and found that the binding sites overlap with regions occupied by histone tails. Post-translation modifications can weaken tail DNA interactions and enhance histone tail dynamics. While these simulations themselves are interesting, I have reservations about their convergence and interpretation.

Response: we would like to thank the reviewer for the concise summary of our study. To address the reviewer's concerns, we have performed additional analyses, increased the simulation time by a factor of 2.5, and revised our explanation of the results. The detailed point-to-point responses are shown below.

2. First, while 26 microseconds may sound impressive and are indeed heroic efforts for atomistic simulations, they do not necessarily support statistical convergence. There are encouraging signs, including the two-fold symmetry shown in Figure SM5 and the correlation between binding free energy from simulations and MM/GBSA calculations.

Response: to further assess the convergence of our simulations, following the reviewer's suggestions, we have compared histone tail binding sites on DNA between the simulation runs starting from different initial configurations. Our results show that these simulations converge on similar tail-DNA binding sites on nucleosomal and linker DNA (Supplementary Table 7). Furthermore, we have significantly increased our simulation time. The simulation time of the long runs was extended from 2.5 microseconds to 4-5 microseconds. The total simulation time has been increased from 26 microseconds to 65 microseconds.

3. However, what concerns me is that the two water models give rise to significant differences in binding and unbinding kinetics. While the authors argue that the OPC model is expected to outperform TIP3P, most references appear to be on water and protein simulations. Is there evidence that supports the OPC model's performance for protein-DNA interactions?

Response: previous work has shown that the TIP3P water model leads to overly compact conformations of intrinsically disordered regions, such as histone tails (IDP), regardless of the underlying gas-phase force-field ¹. The initial signs of histone tail compaction into a globular like particles using TIP3P water were seen in our simulations as well. While a specialized water model TIP4P-D was recently developed to address this defect in simulations of IDPs ¹, a general-purpose water model OPC has been shown to perform well on histone tails ² and IDPs in general ³ (please also see remarks of the second reviewer). Besides proteins, OPC water model has been extensively tested in simulations of RNA ⁴⁻⁶ and DNA ^{7,8}. Most recently, OPC has been applied to simulate the interactions between the linker histone globular domains and DNA in the context of the chromatosomes ⁹. For these reasons we believe that OPC is the best choice in this work, which focuses on the nucleosome. In the revised manuscript, we have compared our results with different experimental studies, even with those which were published after our paper was submitted (please see answers to the points below) and showed that our OPC-based simulations reproduce experiments well.

4. The authors explored different ways to build initial structures for histone tails to avoid biases in their prediction of the binding sites. It would be useful to analyze the difference among results from these initial configurations. Do they more or less converge to the same binding sites, or are there significant differences among them? These analyses could provide more insight into the statistical convergence.

Response: following the reviewer's comments, we have compared the histone tail binding sites on nucleosomal or linker DNA between the simulation models starting from different initial configurations. Our results (added as new Supplementary Fig. 7 and Supplementary Table 14) show significant correlation between the binding site locations of simulations starting from different models, indicating the convergence to similar tail-DNA binding sites on nucleosomal and linker DNA. We have added the explanation in the main text of the paper (Page 5, Line 119-123).

	ModelA	ModelB	ModelC	ModelD
ModelA	1	0.71	0.66	0.58
ModelB	0.71	1	0.71	0.6
ModelC	0.66	0.71	1	0.8
ModelD	0.58	0.6	0.8	1

Supplementary Table 7. Pearson correlation coefficients between the mean number of tail-DNA contacts at each DNA base pair calculated for different nucleosome models.

Supplementary Fig. 14. Mean number of tail-DNA contacts combined from all types of histone tails per model. The numbers are averaged over 1ns frames of simulation runs per model.

5. Second, given that the histone tails cover almost the entire nucleosomal DNA (Figure SM5), it's perhaps not too surprising that there is some overlap between tail and chromatin regulator.

Response: while histone tails can interact with a portion of nucleosomal and linker DNA at some point in time, there is a very limited solvent accessible region of nucleosomal/linker DNA which is covered by tails most of the time in our simulations (Figure 2). In figure 3c (top track), we show the tail-DNA preferred binding modes which are defined as those nucleosomal or linker DNA regions that have more than five contacts, on average. As one can see from Figure 3c (top track), these regions account for only 38% of the total DNA molecular surface area.

6. The implications of this observation are nontrivial. I am not convinced that histone tails serve as roadblocks that must be displaced for protein binding. In particular, histone tails are flexible and adopt rather dynamic binding sites. They could adjust their conformation to accommodate the additional proteins or seek new binding sites.

Response: as we argue in the paper, and as the reviewer points out, the interactions of histone tails are transient and dynamic. However, as we mentioned in our answer to point #5, there are tail-DNA preferred binding modes, some of which have also been previously observed in crosslinking experiments^{10,11}. Histone tails do seek and change DNA binding sites within their preferred binding modes, but as we show in the paper, there is a considerable overlap between the histone tail preferred binding regions and protein binding interfaces on nucleosomal or linker DNA.

Our extensive simulations show the dynamic tails' behavior and characterize their binding and unbinding events. We do not think the term "roadblocks" adequately represents the dynamical model which we describe in the paper. Some nucleosome binding proteins can compete with histone tails on DNA and/or displace histone tails to make them more accessible for recognition by reader domains. Such scenarios have been previously validated in the study of the LSD1-CoREST complex, where the interaction between the SANT2 domain and nucleosomal DNA displaces H3 tails from DNA and facilitates tails' interactions with the LSD1 active site¹². Recent studies also indicated the displacement of H1 C-terminal and H3 tails by binding of HGMN protein to nucleosome¹⁰ and competitive binding between chromatin remodeler ISWI and H3 histone tails⁹. In addition, there are several studies showing that tail-DNA interactions in the context of the nucleosome are quite extensive, leading to the decreased nucleosomal and linker DNA solvent accessibility and affecting the binding of proteins to histone tails or DNA^{9,12-15}.

7. The delta G values in Figure SM4 have no units, but if I interpret that as kcal/mol, then 3kcal/mol for H4 is not a significant number, especially considering that it's for unbinding of the entire tail.

Response: we thank the reviewer for noticing this, we have added the units in Supplementary Fig. 4 (it is Supplementary Fig. 5 in the revised manuscript). The values of binding free energies (ΔG s) for H3 and H4 tail interactions with DNA, estimated by our analyses, are about 2-4 kcal/mol. We should mention that our unbound state retains 10% of its contacts with DNA, so these are lower bound estimates. To address the reviewer's point, we have collected experimentally determined standard state ΔG_0 values of 83 proteins bound to the free DNA from the dbAMEPNI database¹⁶ and in addition 46 proteins bound to nucleosome from a very recent quantitative mass spectrometry study (Supplementary Table 9). Reported standard binding free energies of partners to nucleosomes and DNA are in the range of 8-10 kcal/mol and 3-15 kcal/mol respectively (Supplementary Table 9, Supplementary Fig. 24). Even though these values are for many complexes larger than the binding affinity of the histone tails to nucleosomal DNA, we should note that the probability of binding of tails and partners is controlled by both local concentrations and their binding affinities. The local concentrations of histone tails and nucleosomes are orders of magnitude higher than the concentration of the nucleosome binding partners, which is in the nM to μ M range^{17,18}. Thus, the binding affinity of the nucleosome binding proteins moves into the lower range, where the tail binding has an effect. We have added new analyses and discussion in the revised manuscript (Page 7, line 175 to 184; Page 10, line 264-270; Supplementary Materials, page 2-4).

Supplementary Figure 24. Standard state protein-DNA binding free energies taken from dbAMEPNI database¹⁶. The distribution of experimentally determined binding free energies for 83 DNA binding proteins. The density distribution is smoothed using the gaussian smoothing kernel.

8. These estimations also ignored the protein-tail interactions, which could contribute positively to the binding of nucleosomal proteins.

Response: we agree with the reviewer that tail-protein interactions could contribute positively to the binding of nucleosomal partners. This point is actually shown in our original Figure 5, and in the Discussion section where we describe the multivalent binding model. We have added new analyses of tail-protein binding modes (see new Figure 4) and clarified this in the revised manuscript (page 6-7; Supplementary Materials, Page 2-4). To address this point (also pointed out by reviewer 2), we have performed additional analyses of binding modes between tails and nucleosome binding partners using all 131 available nucleosome complex structures and indeed showed that in many of these complexes, tail-protein interactions contribute positively to the binding of nucleosome proteins, as a reviewer rightfully pointed out. Namely, there are 86 structures where protein binding partner interacts with the nucleosomal/linker DNA, with or without tails involved. In 50% of these structures, binding partners recognize only nucleosomal/linker DNA and do not interact with histone tails (Figure 4). In the second group of 40% of structures, binding partners recognize both histone tails and nucleosomal/linker DNA, but tails are located away from the partner-DNA interfaces. Finally, there are 10% of the structures in the third group, where proteins recognize histone tails and DNA within the close proximity to each other and tails may stay in partially bound states with DNA while interacting with partners.

We further performed electrostatic potential analysis of these 86 PDB structures and results show that in the group of multivalent interactors (Figure 4, right panel), binding partners recognize both histone tails and nucleosomal/linker DNA via two separate patches: acidic (interactions with tails) and basic (interactions with DNA). Here tails contribute positively to the partner's binding to nucleosomes. However, even in these cases, most of the DNA binding interfaces of partners are positively charged and not favorable for histone tail binding.

Finally, please also see the response to the reviewer 2's comment#2 on page 10 where we offered two theoretical models to predict the effective binding free energy of proteins to nucleosomal DNA with tails involved.

Figure 4. Recognition of nucleosomal DNA and histone tails by binding partners depicted via electrostatic potential analysis. Nucleosome complex structures where proteins interact with DNA are classified based on their histone tail binding modes. Positively charged DNA binding interfaces are highlighted for chromatin remodeler INO80 (PDB: 6HTS) and UV-damaged DNA-binding protein (PDB: 6R8Z) where partners do not interact with histone tails in structures. Another two representative examples, chromatin remodeler ISWI (PDB: 6IRO) and polycomb repressive complex 2 (PDB: 6WKR), show the DNA binding interfaces and the partner acidic patches recognized by H3 and H4 tails. Electrostatic potentials are mapped onto the molecular surfaces of nucleosome binding proteins. Blue and red colors indicate the positive and negative electrostatic potentials, and the intensity of the color is scaled with the surface electrostatic potential values. H3 and H4 tails are colored as light blue and green. The molecular surface of nucleosomal and linker DNA is highlighted with cyan color. Core histone regions are not shown for clarity.

9. Numerous prior studies have observed a change in histone tail secondary structure upon DNA binding. It would be useful if the authors could comment on the corresponding changes in their simulations.

Response: following the reviewer's suggestion, we have performed an additional analysis of histone tails' secondary structure and added Supplementary Fig. 6 in the revised manuscript. In our simulations, we sampled histone tail conformations in the context of the full nucleosome, characterized by extensive tail-DNA interactions. Generally, we observed that histone tails remained highly unstructured, and their conformations were described by turn and coil parameters with the exception of the transient alpha helical formation on H3 tail (Supplementary Fig. 6). These

results are consistent with the previous studies^{19,20}. We have added this result to the paper on page 5, line 99-101.

Supplementary Figure 6. Secondary structure propensity per tail residue plotted versus the percentage of frames where it was observed. Different types of secondary structures include Coil(C), Pi-Helix(I), 3-10 Helix(G), Alpha Helix(H), Isolated Bridge(B), Extended configuration(E), and Turn(T).

10. The font size used in many figures is too small and not legible.

Response: thank you for pointing this out. We have increased the front size in all figures.

11. I would be more reserved with the sentence “Overall, it shows a remarkable linear association with $R^2 = 0.9$, pointing to a reasonable conformational sampling of histone tails’ binding performed in our study (Figure 1d)”. The correlation requires fine-tuning the cutoff parameter, and the values for H3 cannot be estimated reliably.

Response: in the revised version, we have significantly extended the simulation time and observed increased correlation coefficients for different cut-off parameters (see Supplementary Fig. 5), pointing to the improved convergence of simulations. There was no fine tuning involved and for all different cutoff values the correlation coefficient is above 0.9. We have also revised the sentence to “Overall, we observe a strong linear association between the histone tail-DNA binding free energies derived from the tail conformational ensemble statistics and MM/GBSA calculations for different values of cut-off parameters (Supplementary Fig. 5). “

12. SHL was never clearly defined.

Response: thank you for pointing this out. SHL (Superhelical Location) has been defined in the revised manuscript.

Reviewer #2

This manuscript describes a set of molecular dynamics simulations that investigated the spatial distribution of full-length histone tails around a nucleosome particle. Probing such distributions using the all-atom MD method has been difficult in the past because of the insufficient simulation time scale and because of the force field artifacts that favored overly compact conformations for intrinsically disordered proteins. The authors have overcome the latter problem by using a custom water model developed by the Onufriev lab. Indeed, instead of seeing a permanent binding of a histone tail to DNA, which is a typical outcome of a simulation performed using either standard AMBER or standard CHARMM force field, the authors observed a multitude of binding and rebinding events, enough to draw statistically significant conclusions on where and for how long the histone tails bind. Armed with that information, the authors find a significant overlap between the DNA regions where the histone tails bind and the DNA regions occupied by DNA binding

proteins in the available crystallographic structures. Interestingly, chemical modifications of histone tail residues were found to alter the histone tail binding in a non-trivial manner. The authors conclude that histone tail binding to DNA can regulate accessibility of the nucleosomal DNA to chromatin remodeling factor and other DNA binding proteins.

1. This is an interesting study that tackles head on an outstanding question in chromatin biophysics. The simulations were carefully designed and carried out in replicate to improve statistical significance of the results. The manuscript text is clearly written, but the figures require improvement. The authors are asked to revise the manuscript to address the following.

Response: we appreciate reviewer's positive comments and constructive suggestions. We have improved the figures and addressed all concerns point by point below.

2. Perhaps the only major criticism of the work is its indirect conjecture on histone tail binding being able to outcompete or reduce protein binding to nucleosomal DNA. The histone displacement model provides a believable route to how protein binding to DNA can proceed despite the DNA being occupied by histone tails. Histone tail might also have a favorable affinity to DNA binding proteins, so it is not at all clear to which degree DNA occlusion translates into modulation of DNA binding affinity. Ideally, the protein binding scenario should have been investigated through free-energy calculations. This reviewer, however, realizes the challenges involved in carrying out such multi-dimensional calculations. A simple resolution could be a theoretical model that, taking the affinity of a DNA binding protein to the cognate DNA fragment and to the histone tails as input, predicts the effective binding constant for the protein. The results of such model could be presented in a new figure, giving the readers an idea about the range of conditions where DNA occlusion by histone tails becomes important.

Response: we agree with the comments that histone tails could have favorable binding affinity to nucleosome binding proteins. It is accounted by the proposed multivalent binding model (Figure 6). In revision, we have further added new discussions and analyses of tail-protein binding modes (Page 6-7; Supplementary Materials, page 2-4). We have performed additional analyses of tail-protein binding modes using all available 131 nucleosome complex structures and added a new Figure 4 (details are also shown in the response to reviewer 1's comment#8, page 6). Our results

indeed show that in many of these complexes, tail-protein interactions contribute positively to the binding of nucleosome proteins, as a reviewer rightfully pointed out. We further performed electrostatic potential analysis of these PDB structures and showed that in the group of multivalent interactors (Figure 4, right panel), binding partners recognize both histone tails and nucleosomal/linker DNA via two separate patches: acidic (interactions with tails) and basic (interactions with DNA). Here tails contribute positively to the partner's binding to nucleosomes.

Following the reviewer 2's suggestions, we further offer theoretical models to predict the effective binding free energy of protein to nucleosomal DNA with tails involved. These new results and discussions have been added in the revised manuscript (Page 6-7; Supplementary Materials, page 2-4).

First, we predict the effective dissociation constant of a partner bound to nucleosome using thermodynamic cycle (Supplementary Fig. 16). As one can see from the thermodynamic cycle and the predicted value, tail-protein interactions may contribute positively to the overall binding between a partner and nucleosomes while tail-DNA interactions contribute negatively.

If we do not consider partner-tail interactions, in case they are highly unfavorable for binding each other, then we can calculate the selectivity constant K_{pt} to describe the binding of a partner to a tail-DNA complex (in a nucleosome). The results imply that in cases where histone tails and partner occupy overlapping regions on nucleosomal DNA (Figure 3d) – it can considerably decrease the binding of partners to nucleosomes.

Figure 4. Recognition of nucleosomal DNA and histone tails by binding partners depicted via electrostatic potential analysis. Nucleosome complex structures where proteins interact with DNA are classified based on their histone tail binding modes. Positively charged DNA binding interfaces are highlighted for chromatin remodeler INO80 (PDB: 6HTS) and UV-damaged DNA-binding protein (PDB: 6R8Z) where partners do not interact with histone tails in structures. Another two representative examples, chromatin remodeler ISWI (PDB: 6IRO) and polycomb repressive complex 2 (PDB: 6WKR), show the DNA binding interfaces and the partner acidic patches recognized by H3 and H4 tails. Electrostatic potentials are mapped onto the molecular surfaces of nucleosome binding proteins. Blue and red colors indicate the positive and negative electrostatic potentials, and the intensity of the color is scaled with the surface electrostatic potential values. H3 and H4 tails are colored as light blue and green. The molecular surface of nucleosomal and linker DNA is highlighted with cyan color. Core histone regions are not shown for clarity.

Supplementary Figure 16. Thermodynamic cycle to estimate the binding free energy of protein to tail-DNA complex. $\Delta G(\text{Pro-Tail-DNA})$, $\Delta G(\text{Tail-DNA})$, $\Delta G(\text{Pro-DNA})$, and $\Delta G(\text{Tail-Pro})$ represent binding free energies of protein to tail-DNA complex, histone tail to free DNA, protein to free DNA, and histone tail to protein-DNA complex.

3. The 250 ns simulation time scale is a bit short for 2021. Any chance all simulations could be run for 2.5 microsecond? If not, consider extending to what is technically feasible while working on the revisions.

Response: we regret that all 22 simulations cannot be extended to 2.5 microsecond due to limitations of computational resources. We have extended the simulation time of all short runs from 200ns to 800ns, and the total simulation time for all the runs has been extensively increased from 26 microseconds to 65 microseconds.

4. The authors accumulated significant amount of data on tail binding to DNA. Can the authors say anything general on how the binding depends on the tail length, charge and/or hydrophobicity?

Response: following the reviewer's suggestions, we have added a discussion of how tail binding depends on the tail length, charge, and/or hydrophobicity in the revised manuscript (Page 5, line 105-109; Page 9, line 246-252).

5. Previous simulations and experiments found AT-content to correlate with the affinity of poly-lysine peptides to DNA [Nucleic Acids Research 46: 9401]. Do the authors see any sequence/AT content preference for histone tail binding?

Response: we thank the reviewer for pointing out this interesting paper which we cited in the revised manuscript. As a reviewer suggested, we have analyzed the interactions of histone tails with AT and GC base pairs using the simulation trajectories. However, compared to the previous study ²¹, we do not observe significant differences between histone tails' interactions with AT versus GC nucleotide base pairs.

Figure 1. Histone tail mean contact numbers with AT and GC nucleotide content. Each point represents a mean contact number between one DNA base pair of a given type and histone tail residues calculated from simulation trajectories. a) mean contact number between histone tails and DNA for different base pairs. b) mean contacts number between histone tail lysine residues and DNA for different nucleotide base pairs.

6. Please clarify the initial conformation used for the simulations of the modified tails systems.

Response: thank you for pointing this out. We have used the model D (histone tails were extended into the solvent symmetrically oriented with respect to the dyad axis) for the simulations of the modified tail systems. We have clarified this in the revised manuscript (Page 14, line 428-429).

7. The effect of modifications on tail binding is striking. What is lacking is some kind of statistical measure of its significance. Ideally, these simulations should be run longer.

Response: to address the reviewer's point, we have extended the total simulation time of modified histone tails from 9 microseconds to 24 microseconds and added new analyses in the revised version (Supplementary Fig. 23). We have evaluated the significance of tail binding

changes upon modifications. First, we have calculated the change of the mean number of contacts per DNA base pair and its standard errors (Supplementary Fig. 23). To find out more about how modifications affect the overall tail-DNA interactions, for each modification type we have applied the t-test (Supplementary Fig. 23). We have revised the text accordingly (page 8, line 208-215).

Supplementary Figure 23. The change of the mean number of contacts between histone tails and nucleosomal and linker DNA per base pair upon modifications. For each modification type, the change of the mean number of contacts per DNA base pair is calculated as the mean number of contacts between DNA and modified tails minus the mean number of contacts between DNA and unmodified tails. The reported values are averaged and the error bars represent the standard errors of the mean for modified tails calculated from independent simulation runs. The t-test has been performed for analyzing the statistical significance of changes of the full tail-DNA contacts upon modifications. The null hypothesis is that the mean change of tail-DNA contacts upon modifications is zero. The alternative hypothesis is that tail modification decreases the overall mean contact number between tail and DNA. Table lists p-values, if p-value is less than 0.05 the change is considered to be significant for the full tail. For each DNA site significant changes occur if the error bars do not extend beyond zero.

8. Page 12, first sentence. Panel f is not referenced before Panel b.

Response: thank you for pointing this out. We have changed the order of the Panel f in the revised Figure 3.

9. Page 12, it would be good to specify typical Delta G values for protein binding to DNA here and compare them to tail binding values.

Response: the values of binding free energies (ΔG s) for H3 and H4 tail interactions with DNA, estimated by our analyses, are about 2-4 kcal/mol. We should mention that our unbound state retains 10% with DNA so it is the lower bound estimate. Following the reviewer 2's suggestions, we have collected the experimentally determined standard state ΔG_0 values of 83 proteins bound to the free DNA from the dbAMEPNI database¹⁶ and in addition 46 proteins bound to nucleosome from a very recent quantitative mass spectrometry study (Supplementary Table 9). Reported standard binding free energies of partners to nucleosomes and DNA are in the range of 8-10 kcal/mol and 3-15 kcal/mol respectively (Supplementary Table 9, Supplementary Fig. 24). Even though these values are for many complexes larger than the binding affinity of the histone tails to nucleosomal DNA, we should note that the probability of binding of tails and partners is controlled by both local concentrations and their binding affinities. The local concentrations of histone tails and nucleosomes are orders of magnitude higher than the concentration of the nucleosome binding partners, which is in the nM to μ M range^{17,18}. Thus, the binding affinity of the nucleosome binding proteins moves into the lower range, where the tail binding has an effect. We have added new analyses and discussion in the revised manuscript (Page7, line175 to 184; Page 10, line 264-270; Supplementary Materials, page 2-4).

Supplementary Figure 24. Standard state protein-DNA binding free energies taken from dbAMEPNI database¹⁶. The distribution of experimentally determined binding free energies for 83 DNA binding proteins. The density distribution is smoothed using the gaussian smoothing kernel.

10. Lines 403-405: This sentence implies that histone tails facilitate binding of chromatin factor, in contrast to the next sentence. Maybe, to emphasize that fact better, start the next sentence with “In contrast”?

Response: thank you for pointing this out. This sentence actually indicates that the binding of proteins to histone tails can be largely weakened by the tail-DNA interactions within the nucleosome. In the revised manuscript, we have rephrased the sentence to “*Indeed, recent study has shown that the PHD fingers of CHD4 bind up to ten-fold tighter to histone tail peptides compared to binding to the tails in the context of the nucleosome.*”

11. Line 413 starts with “Our second prediction”. What was the first one? The previous paragraph does not explicitly describe one.

Response: thank you for pointing this out. We have removed the “*Our second prediction*” in the revised manuscript.

Line 426: Please replace “them” with either “tails” or “modes”, depending on the intended meaning.

Response: we have replaced “them” with either “tails” or “modes” in the following sentence: “*It makes tails more accessible for the recognition by reader domains (Figure 6).*”

12. Figure 1 and most of the figures. The axis labels are way too small, almost microscopic. The figures appear pixelated (a pdf to MS word conversion, likely), difficult to extract quantitative information.

Response: thank you for pointing this out. We have separated Figure 1 into two parts and Figure 1f has been moved to Supplementary Fig. 4. In addition, we have increased the font size of axis labels for all figures in the revised manuscript.

13. Figure 1d: The MM/GBSA free energies are way too large (by magnitude) to be realistic. The biotin-streptavidin binding energy of -18kcal/mol guarantees almost irreversible

binding. A binding energy of -50kcal/mol signifies a permanent attachment. Maybe remove MM/GBSA results?

Response: we agree that the MM/GBSA calculations usually overestimate the binding free energy values, and this has already been shown in many studies. Therefore, in revisions our results and conclusions only emphasize the relative instead of the absolute values obtained from MM/GBSA calculations. We removed the MM/GBSA results from Figure 1d in the revised manuscript.

14. Figure 2a, c and d: Define SHL

Response: thank you for pointing this out. SHL (Superhelical Location) has been defined in the revised figure.

15. Figure 2d: The Y axis should probably be “fraction” not “percentage”

Response: we have replaced “percentage” to “fraction” in the revised figure.

16. Figure 3b is confusing. The top panel does not have any axes or units. The title above panels does not help.

Response: thank you for pointing this out. We have added the axis in the revised figure and revised the figure caption.

17. Figure 3c: The Y axis (density) should have some kind of units.

Response: thank you for pointing this out. The Y axis (density) represents the smoothed kernel density estimates of structure number and thus should be unitless. We have clarified this in the figure caption.

18. Figure 3 caption. In b, start the caption with “Crystallographic analysis of ... “ or something similar.

Response: as the reviewer suggested, we have revised the figure caption.

19. Figure 3 caption. In c, the caption mentions smoothing with a kernel, what were the parameters of the kernel?

Response: the density plot was smoothed with a gaussian kernel function with the default parameter and bandwidth (implemented in `geom_density()` in R). We have clarified this in the revised figure caption.

20. Figure 4: Axes are missing in panel a

Response: thank you for pointing this out. We have added the axis and labels in the revised figure (Supplementary Fig. 17 in the revised manuscript).

Reviewer #3:

Binding or regulatory to nucleosomes is modulated by dynamic histone tails
By Pachenko et al.

1. This comprehensive simulation work on effects of PTM on NCP histone tail modifications is significant. My only concern is that the length of simulation for PTM and oncogenic mutations (2000 ns) may not allow for the relaxation of the tail positions. Computational results for PTM effects on binding affinity of tail are reported and compared to experimentally reported results. This article should be considered for publication in Nature Communications after same major concerns are addressed:

Response: we appreciate reviewer's positive comments and constructive suggestions. Per reviewer's request, we have extensively extended the simulation time to improve the convergence. The total simulation time for modified tails has been increased from 9 microseconds to 24 microseconds. To further assess the convergence of our simulations, we have compared histone tail binding sites on DNA between the simulation runs starting from different initial configurations. Our results show that these simulations converge on similar tail-DNA binding sites on nucleosomal and linker DNA (please see figures and tables above in response to reviewer1's comments#4). We further addressed the concerns point by point below.

2. Binding and unbinding behavior of the tails reported here is entirely force field dependent. This is quite interesting. What validation is there that the Amber OL15 force fields more accurately reproduce timescales of experimental binding/unbinding?

Response: binding and unbinding behavior of histone tails is generally more dependent on the water models and less dependent on the force fields. Previous work has shown that the TIP3P water model tends to predict overly compact conformations of intrinsically disordered proteins IDPs, regardless of the underlying gas-phase forcefield ¹. The general-purpose water model OPC has been shown to perform well on histone tails ² and IDPs in general ³ and, most recently, was directly applied to simulate the interactions between the linker histone globular domains and DNA in the context of the chromatosomes⁹. The initial signs of histone tail nonviable compaction into a globular like particles have been shown in our simulations with TIP3P model as well, that's why we decided to choose the OPC model.

Critically, we have compared our results to recent experimental data and added a discussion in the revised manuscript (page 9, line 241-252). A large body of experimental evidence points to the high level of conformational dynamics of histone tails, with the dynamic transitions of conformations on the order of sub-microseconds ^{13,22,23}. This is generally consistent with our observed highly dynamic tail behavior in the simulations with the OPC water model. A very recent FRET study shows that H3 tails have multiple interaction modes with nucleosomal or linker DNA, with conformational transitions from compact state to extended state (may correspond to tail binding/unbinding with DNA) taking place on micro- to millisecond timescales ²⁴. Our results show that compared to other tails, H3 tail has the longest residence time on DNA, on the order of microseconds or longer. Due to limitations on the timescales accessible to atomistic simulations, it is still impossible to assess the dynamics on the order of milliseconds for such large systems as nucleosomes. Therefore, our results and conclusions in the revised manuscript mainly emphasize the relative differences of the tail binding behavior among different tail types instead of the absolute values.

3. More discussion concerning significance of PTM on larger scale chromatin structure and phase behavior should be reported. All effects of PTM are discussed on single nucleosome level. How would this affect nucleosome-nucleosome interactions, etc? How would PTM affect the dynamics?

Response: thank you for the constructive suggestions, we have added the discussion in the revised version (Page 11, line 304-312).

4. Minor Concern: Figure SM4 Unbinding is spelled incorrectly. Figure 1 contains too many panels and the captions are exceedingly hard to read. This figure should be split up and a portion can be moved to the SI.

Response: thank you for pointing this out. We have corrected the misspelling in Supplementary Fig. 4 (it is now Supplementary Fig. 5 in the revised manuscript). In addition, we have splitted Figure 1 into two parts and Figure 1f has been moved to Supplementary Fig. 4.

Reference

- 1 Piana, S., Donchev, A. G., Robustelli, P. & Shaw, D. E. Water dispersion interactions strongly influence simulated structural properties of disordered protein states. *J Phys Chem B* **119**, 5113-5123, doi:10.1021/jp508971m (2015).
- 2 Shabane, P. S. & Onufriev, A. V. Significant compaction of H4 histone tail upon charge neutralization by acetylation and its mimics, possible effects on chromatin structure. *J Mol Biol*, 166683, doi:10.1016/j.jmb.2020.10.017 (2020).
- 3 Shabane, P. S., Izadi, S. & Onufriev, A. V. General Purpose Water Model Can Improve Atomistic Simulations of Intrinsically Disordered Proteins. *J Chem Theory Comput* **15**, 2620-2634, doi:10.1021/acs.jctc.8b01123 (2019).
- 4 Bergonzo, C. & Cheatham, T. E., 3rd. Improved Force Field Parameters Lead to a Better Description of RNA Structure. *J Chem Theory Comput* **11**, 3969-3972, doi:10.1021/acs.jctc.5b00444 (2015).
- 5 Dans, P. D. *et al.* Modeling, Simulations, and Bioinformatics at the Service of RNA Structure. *Chem* **5**, 51-73, doi:10.1016/j.chempr.2018.09.015 (2019).
- 6 Kuhrova, P. *et al.* Improving the Performance of the Amber RNA Force Field by Tuning the Hydrogen-Bonding Interactions. *J Chem Theory Comput* **15**, 3288-3305, doi:10.1021/acs.jctc.8b00955 (2019).
- 7 Galindo-Murillo, R. *et al.* Assessing the Current State of Amber Force Field Modifications for DNA. *J Chem Theory Comput* **12**, 4114-4127, doi:10.1021/acs.jctc.6b00186 (2016).
- 8 Yang, C., Kulkarni, M., Lim, M. & Pak, Y. Insilico direct folding of thrombin-binding aptamer G-quadruplex at all-atom level. *Nucleic Acids Res* **45**, 12648-12656, doi:10.1093/nar/gkx1079 (2017).
- 9 Zhou, B. R. *et al.* Distinct Structures and Dynamics of Chromatosomes with Different Human Linker Histone Isoforms. *Mol Cell* **81**, 166-182 e166, doi:10.1016/j.molcel.2020.10.038 (2021).
- 10 Murphy, K. J. *et al.* HMGN1 and 2 remodel core and linker histone tail domains within chromatin. *Nucleic Acids Res* **45**, 9917-9930, doi:10.1093/nar/gkx579 (2017).
- 11 Lee, K. M. & Hayes, J. J. The N-terminal tail of histone H2A binds to two distinct sites within the nucleosome core. *Proc Natl Acad Sci U S A* **94**, 8959-8964, doi:10.1073/pnas.94.17.8959 (1997).
- 12 Pilotto, S. *et al.* Interplay among nucleosomal DNA, histone tails, and corepressor CoREST underlies LSD1-mediated H3 demethylation. *Proc Natl Acad Sci U S A* **112**, 2752-2757, doi:10.1073/pnas.1419468112 (2015).
- 13 Morrison, E. A., Bowerman, S., Sylvers, K. L., Wereszczynski, J. & Musselman, C. A. The conformation of the histone H3 tail inhibits association of the BPTF PHD finger with the nucleosome. *Elife* **7**, doi:10.7554/eLife.31481 (2018).
- 14 Gatchalian, J. *et al.* Accessibility of the histone H3 tail in the nucleosome for binding of paired readers. *Nat Commun* **8**, 1489, doi:10.1038/s41467-017-01598-x (2017).
- 15 Skrajna, A. *et al.* Comprehensive nucleosome interactome screen establishes fundamental principles of nucleosome binding. *Nucleic Acids Res*, doi:10.1093/nar/gkaa544 (2020).
- 16 Liu, L. *et al.* dbAMEPNI: a database of alanine mutagenic effects for protein-nucleic acid interactions. *Database (Oxford)* **2018**, doi:10.1093/database/bay034 (2018).
- 17 Milo, R. & Phillips, R. *Cell biology by the numbers*. (Garland Science, 2015).
- 18 Langst, G. & Manlyte, L. Chromatin Remodelers: From Function to Dysfunction. *Genes (Basel)* **6**, 299-324, doi:10.3390/genes6020299 (2015).
- 19 Erler, J. *et al.* The role of histone tails in the nucleosome: a computational study. *Biophys J* **107**, 2911-2922, doi:10.1016/j.bpj.2014.10.065 (2014).
- 20 Biswas, M., Voltz, K., Smith, J. C. & Langowski, J. Role of histone tails in structural stability of the nucleosome. *PLoS Comput Biol* **7**, e1002279, doi:10.1371/journal.pcbi.1002279 (2011).
- 21 Kang, H. *et al.* Sequence-dependent DNA condensation as a driving force of DNA phase separation. *Nucleic Acids Res* **46**, 9401-9413, doi:10.1093/nar/gky639 (2018).

- 22 Zhou, B. R. *et al.* Histone H4 K16Q mutation, an acetylation mimic, causes structural disorder of its N-terminal basic patch in the nucleosome. *J Mol Biol* **421**, 30-37, doi:10.1016/j.jmb.2012.04.032 (2012).
- 23 Ghoneim, M., Fuchs, H. A. & Musselman, C. A. Histone Tail Conformations: A Fuzzy Affair with DNA. *Trends Biochem Sci*, doi:10.1016/j.tibs.2020.12.012 (2021).
- 24 Lehmann, K. *et al.* Dynamics of the nucleosomal histone H3 N-terminal tail revealed by high precision single-molecule FRET. *Nucleic Acids Res* **48**, 1551-1571, doi:10.1093/nar/gkz1186 (2020).

REVIEWERS' COMMENTS

Reviewer #1 (Remarks to the Author):

The authors have successfully addressed all my comments and I now support its publication.

Reviewer #2 (Remarks to the Author):

The authors have addressed all the issues brought about in the previous round of review. The manuscript has been considerably improved with regard to both the statistical significance of the results and their scholarly presentation. The manuscript can be accepted to publication in its present form.

Aleksei Aksimentiev

Reviewer #3 (Remarks to the Author):

The authors have satisfactorily answered my concerns regarding this manuscript. The publication should be accepted in Nature Communications.